



# Attribution of precipitation to cyclones and fronts over Europe in a kilometer-scale regional climate simulation

Stefan Rüdisühli[1], Michael Sprenger[1], David Leutwyler[2], Christoph Schär[1], and Heini Wernli[1]

[1]Institute for Atmospheric and Climate Science, ETH Zurich, Switzerland
[2]Max Planck Institute for Meteorology, Hamburg, Germany

**Correspondence:** Stefan Rüdisühli (stefan.ruedisuehli@env.ethz.ch)

**Abstract.**

This study presents a detailed analysis of the climatological distribution of precipitation in relation to cyclones and fronts over Europe for the nine-year period 2000–2008. The analysis uses hourly output of a COSMO (Consortium for Small-scale Modeling) model simulation with 2.2 km grid spacing and resolved deep convection. Cyclones and fronts are identified as two-dimensional features in 850 hPa geopotential, equivalent potential temperature, and wind fields, and subsequently tracked over time based on feature overlap and size. Thermal heat lows and local thermal fronts are removed based on track properties. This data set then serves to define seven mutually exclusive precipitation components: high-pressure (e.g., summer convection), cyclonic (near cyclone center), cold-frontal, warm-frontal, collocated (e.g., occlusion area), far-frontal, and residual. The approach is illustrated with two case studies with contrasting precipitation characteristics. The climatological analysis for the nine-year period shows that frontal precipitation peaks in fall and winter over the eastern North Atlantic, with cold frontal precipitation also being crucial year-round near the Alps; cyclonic precipitation is largest over the North Atlantic (especially in summer) and in the northern Mediterranean (except in summer); high-pressure precipitation occurs almost exclusively over land and primarily in summer; and the residual contributions uniformly amount to about 20 % in all seasons. Considering heavy precipitation events (defined based on the local $99.9^{th}$ percentile) reveals that high-pressure precipitation dominates in summer over the continent; cold fronts produce much more heavy precipitation than warm fronts; and cyclones contribute substantially, especially in the Mediterranean in fall through spring and in Northern Europe in summer.

## 1 Introduction

Precipitation is one of the most central meteorological variables and, therefore, huge efforts have been invested in compiling regional and global precipitation climatologies from surface station measurements, remote-sensing data, and combinations thereof (e.g., Xie and Arkin, 1997; Frei and Schär, 1998; Adler et al., 2003; Sun et al., 2018; Isotta et al., 2014). Such climatologies, with typically monthly time resolution, serve to characterize the spatial patterns, seasonal cycle, and interannual



variability of precipitation, and they are valuable for strategic decisions in different socio-economic sectors (e.g., water management, agriculture, hydropower generation). Long-term climatologies reveal large interannual variability and trends (e.g.,

Klein Tank and Können, 2003; Zolina et al., 2010). Among the most important questions for future climate change is how a warmer climate will affect precipitation and its climatological distribution, seasonality, interannual variability, and the occurrence of extreme events. In the global mean, precipitation is expected to increase at a rate of 2 % per degree global-mean warming, but changes in short-term precipitation are likely to occur at much faster rates (Trenberth, 1999; Held and Soden, 2006; Schneider et al., 2010). In the last decade, huge progress has been made in realistically simulating the hydrological cycle

with high-resolution climate models, including the spatial distribution of precipitation, its diurnal cycle, and the statistics of extreme events (e.g., Hohenegger et al., 2008; Kendon et al., 2012; Ban et al., 2014, 2015; Prein et al., 2015; Clark et al., 2016; Keller et al., 2016; Leutwyler et al., 2017). A major part of this progress is due to the step-change of simulating deep convection explicitly instead of using a parameterized representation. In their systematic comparison of climate model simulations with parameterized or explicit convection, Prein et al. (2015) found that "Improvements [when using explicit convection] are

evident mostly for climate statistics related to deep convection, mountainous regions, or extreme events."

An important aspect of understanding the precipitation climatology and, eventually, its sensitivity to climate change, is the linkage of precipitation to synoptic-scale weather systems. As outlined below, high-pressure systems, extratropical cyclones, fronts, orography, and their interactions contribute essentially to the formation of precipitation in the mid-latitudes, including extreme events related to deep convection. Research in this area has so far mainly followed two strands: (i) detailed investiga-

tions of specific high-impact precipitation events, their large-scale precursors and mesoscale dynamics (e.g., Buzzi et al., 1998; Massacand et al., 1998; Zängl, 2007; Stucki et al., 2012; Grams et al., 2014; Piaget et al., 2015); and (ii) global climatologies to quantify the relevance of cyclones, fronts, warm conveyor belts, and atmospheric rivers for total and/or extreme precipitation (e.g., Catto et al., 2012; Pfahl and Wernli, 2012; Catto and Pfahl, 2013; Lavers and Villarini, 2013; Pfahl et al., 2014; Hénin et al., 2019). However, most climatological studies on the relationship between precipitation and synoptic weather systems are

based on global reanalyses with a typical resolution of 100 km in space and 6 h in time. Such a resolution is clearly inadequate to capture phenomena like short-duration convective precipitation events or the complex interplay between fronts, steep topography, and precipitation. In addition, a distinction between convective and stratiform precipitation is challenging at such resolutions. While some models distinguish stratiform (explicit) and convective (parameterized) precipitation, the convective fraction strongly depends on the model (Fischer et al., 2015, see their Fig. 3).

This study aims at filling these gaps by using high-resolution data to quantify the co-occurrence of precipitation and a set of weather systems over Europe in the present-day climate. To this end, kilometer-scale climate simulations with explicit convection provide the ideal data base to perform such a methodologically and computationally demanding analysis. The following paragraphs provide a concise summary of the link between precipitation and frontal cyclones, summarize how this link has been quantified in previous studies, explain in more detail the usefulness of high-resolution climate simulations for

studying this link on climatological timescales, and outline the specific objectives of this study.

Explaining the surface precipitation pattern has been a major aspect of the Norwegian cyclone model, introduced almost a century ago by Bjerknes and Solberg (1922). They realized that most cyclones are associated with a warm front, which



slopes gently forward with height and produces widespread, rather uniform precipitation of moderate intensity; followed by a cold front, which is steeper, slopes rearward with height, and produces much more intense but less widespread precipitation (Bjerknes, 1919). In the time since, several aspects of the original Norwegian model have been revised, and new features of extratropical cyclones have been introduced. On the large scale, an important addition has been the concept of characteristic airstreams, among them the warm conveyor belt, a warm and moist airstream that ascends along and ahead of the cold front and overruns the warm front, all the while producing large amounts of precipitation (Harrold, 1973; Pfahl et al., 2014). Recent studies suggest that embedded convection can occur within the mostly stratiform cloud band formed by this airstream, leading to intense peaks in surface precipitation (Neiman et al., 1993; Flaounas et al., 2016; Oertel et al., 2019). Observational studies have also revealed complex mesoscale structures in and around the large-scale frontal precipitation areas. In the vicinity of the warm front, there may be about 50 km-wide intense warm-frontal rainbands (e.g., Herzegh and Hobbs, 1980; Colle et al., 2017). In the comparatively dry warm sector, isolated mesoscale precipitation areas can occur; 10 km to 100 km in size, they are triggered by large-scale ascent and topography (Browning et al., 1974). Along the cold front, mesoscale systems such as rainbands, squall lines, or thunderstorms can develop (Hobbs et al., 1980; Browning and Roberts, 1996; Cotton et al., 2011). In the cold sector behind a frontal cyclone, where cold advection and large-scale subsidence prevail, shallow-convective shower cells typically produce intermittent precipitation of light to moderate intensity over a large area (e.g., Weusthoff and Hauf, 2008; Posselt et al., 2008). This very brief summary clearly indicates the complex and rich mesoscale substructures of surface precipitation in extratropical cyclones. In addition, isolated deep convection and the formation of mesoscale convective systems also frequently occur within surface anticyclones and in situations with weak sea-level pressure gradients (e.g., Trentmann et al., 2009; Langhans et al., 2013).

In the past, a variety of approaches have been used to quantify the occurrence of precipitation in cyclones and across fronts. For surface cyclones (and anticyclones), such an attribution is methodologically straightforward once they have been identified as two-dimensional features, for instance bounded by closed sea-level pressure contours (Pfahl and Wernli, 2012). For fronts, such an attribution is less straightforward, because objective frontal identification can be difficult, and because fronts are typically identified as one-dimensional line objects. Classically, cross-frontal profiles of precipitation have been derived from station measurements for single events or as multi-annual composites of frontal passages, for instance in Berlin (Fraedrich et al., 1986), Munich (Hoinka, 1985), and Helsinki (Sinclair, 2013). While such studies can capture the full natural variability of fronts at a certain location, it is difficult to generalize the results to other locations or to larger areas. Studying frontal precipitation climatologically over large areas requires gridded precipitation and temperature data, automated front detection, and precipitation attribution. While such methods are in principle objective, choosing specific approaches and configurations involves many ultimately subjective choices. Lacking a universally accepted definition of fronts, it is not inherently clear how to identify them, and consequently, many different approaches exist, as discussed in detail by Schemm et al. (2018) and Thomas and Schultz (2019). Another subjective choice is involved when attributing precipitation to a front within a certain distance, which might also depend on the resolution of the available data sets. For instance, Catto et al. (2012) used a 5°-wide search box to attribute precipitation from a global measurement data set to fronts based on reanalysis fields on a coarse 2.5° grid. Also



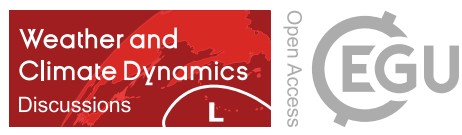

using reanalysis data, Papritz et al. (2014) first identified coherent precipitation objects and then attributed them to objectively identified cyclones and fronts based on overlap criteria.

This brief summary of approaches to attribute precipitation to weather systems – in particular fronts – together with the
mesoscale characteristics discussed above, illustrate a range of challenges: (i) Precipitation data with high spatial and temporal resolution is essential to capture (embedded) convective events (ideally, 1 km and 1 h); (ii) high-resolution fields of (equivalent) potential temperature are required to accurately determine the position and evolution of fronts, in particular near orography; (iii) data with homogeneous quality must be available, ideally on a continental-scale domain and for at least a decade, in order to compile robust climatologies; and (iv) computationally efficient algorithms need to be developed to objectively identify
fronts, cyclones, and anticyclones, as well as for automatic attribution of precipitation to these features. Currently, purely observational data sets hardly meet requirements (i–iii), although hourly gridded precipitation data recently became available from satellites (Sun et al., 2018). Reanalyses might become an option, given that global fields from ERA5 reanalysis (Hersbach et al., 2020) by the European Centre for Medium-Range Weather Forecasts (ECMWF) are available with hourly resolution on a 30 km grid, and regional reanalyses, e.g., the German product COSMO-REA2 (Wahl et al., 2017), exist with a 2 km grid
spacing and high temporal resolution. Currently, however, such high-resolution regional analyses are limited to sub-continental domains, which makes it difficult to meaningfully identify cyclones and fronts (as discussed below). For now, the best option is to use data from continental-scale decadal climate simulations performed with a high-resolution model with explicit deep convection. Recently, such simulations became feasible thanks to a major investment in porting the COSMO (Consortium for Small-scale Modeling) weather and climate prediction model to GPU architectures (Fuhrer et al., 2014; Leutwyler et al., 2016;
Schär et al., 2020). For this study, output from a European-scale COSMO simulation for a 10-year present-day climate period – with 2.2 km grid spacing, explicit convection, and hourly output – will be used to perform a detailed climatological attribution of simulated precipitation to relevant weather systems. The main advantages of this approach are the consistency between the high-resolution data set of surface precipitation and those of all the other meteorological fields required for identifying weather systems; the comparatively large domain covering most of Europe (see Fig. 1); and the explicit treatment of deep convection,
leading to an improved realism in representing the diurnal cycle of summertime precipitation and extreme events. The drawback of using climate model data is that, despite using reanalyses as lateral boundary conditions, individual precipitation systems in the interior of the domain may develop differently in the simulation compared to reality. Therefore, it does not allow for an accurate precipitation attribution for a specific event in the simulation period; instead, it leads to a detailed climatological analysis of the role of anticyclones, cyclones, and fronts for total and heavy precipitation in Europe, separately for each season.
The main objectives of this study are to:

1. develop algorithms that can meaningfully and efficiently identify and track surface high-pressure systems, cyclones, and fronts in the kilometer-scale climate simulation, and robustly attribute hourly precipitation to these weather systems;

2. quantify the contributions to total precipitation of cold fronts, warm fronts, cyclone centers, and high-pressure systems, and investigate the geographical and seasonal variability of this attribution; and



3. as above, but for heavy precipitation, defined annually and seasonally as hourly precipitation exceeding the respective grid-point-specific 99.9[th] all-hour percentile.

In Sec. 2, we introduce the data set and the methodology; in Sec. 3, we demonstrate our approach with two case studies; in Sec. 4, we present climatological results from the precipitation attribution; and in Sec. 5, we summarize the main findings.

## 2   Data and Methods

### 2.1   Simulation and Field Preprocessing

We use hourly output from a ten-year regional climate simulation (1 January 1999 to 31 December 2008) with explicit deep convection over Europe, performed with a GPU-enabled prototype of the COSMO model (version 4.19) (Fuhrer et al., 2014). A detailed description and evaluation of the simulation along with the detailed model setup is found in Leutwyler et al. (2016, 2017); Hentgen et al. (2019). We only analyze the nine-year period from 1 January 2000 to 31 December 2008 because not

all fields necessary for our analysis have been archived during the first few months of the simulation. The domains of the nested COSMO simulations with 12 km and 2.2 km grid spacing are shown in Fig. 1, together with the analysis domain of the high-resolution nest and the model topography. The analysis domain corresponds to the full computational domain minus the grid points that are affected by the boundary relaxation. The COSMO simulation in the high-resolution nest, with a horizontal grid spacing of 2.2 km, has been performed on a $1536 \times 1536 \times 60$ grid. At the boundaries, it is driven by one-way nesting in

a COSMO simulation with a horizontal grid spacing of 12 km on a $355 \times 355 \times 60$ grid. The domain of this coarser simulation is approximately 500 km larger in every direction than that of the nested simulation. In the 12 km simulation, deep convection is parameterized with an adapted version of the Tiedtke mass flux scheme (Tiedtke, 1989). The coarser COSMO simulation in turn is driven at the boundaries by global ECMWF Interim Reanalysis data available on a $1°$ grid (Dee et al., 2011).

    The high spatial resolution of the 2.2 km simulation presents some challenges for the objective identification of cyclones

and fronts. While the domain of the 2.2 km simulation is large given its horizontal resolution, it is still relatively small with respect to these synoptic systems. This causes problems, for instance, when a large-scale North Atlantic cyclone enters the domain from the west, because our algorithm cannot robustly identify cyclone features close to the boundaries, especially if their center (defined as the local pressure minimum) has yet to enter the domain. In addition, the high horizontal resolution can present challenges for our frontal identification algorithm, which is based on horizontal gradients (see below). From a

technical perspective, the driving 12 km simulation thus at first glance appears to be more suitable to identify cyclones and fronts. However, this ignores that small-scale processes resolved in the nested 2.2 km simulation influence the fronts and cyclones, as evidenced by their sometimes substantially different development in the two simulations in terms of their exact size and location – especially far from the lateral boundaries and in the Mediterranean. These differences make it impossible to simply base the feature identification on the 12 km simulation. In order to exploit the advantages of both simulations, the

2.2 km and 12 km data are merged in the following three-step procedure:





1. Interpolate the 2.2 km fields onto the 12 km grid. This retains the exact position and extent of the cyclones and fronts in the 2.2 km simulation while increasing the signal-to-noise ratio to the level of the 12 km simulation.

2. Paste these into the 12 km fields to obtain hybrids comprised of 2.2 km simulation data in the center and 12 km simulation data beyond the boundaries of the inner nest.

3. Introduce a blending zone along the boundaries in the inner domain with a smooth transition from the 2.2 km data to the 12 km data. It extends 50 coarse grid points ($\sim 60$ km) into the inner domain and is based on the logistic function $1/(1 + \exp{-k \times x})$ with $k = 0.8$.

The resulting hybrid fields reside on the grid of the 12 km simulation and thus benefit from its large domain and relatively low noise level, while being meteorologically consistent with the 2.2 km simulation within the analysis domain in the inner
nest. We use them to identify cyclones (Sec. 2.2) and fronts (Sec. 2.3). Before conducting the precipitation attribution analysis (Sec. 2.5), however, the features are interpolated back onto the original 2.2 km grid.

## 2.2 Cyclones

The cyclone identification is based on the approach by Wernli and Schwierz (2006), who identified cyclones as two-dimensional features defined by closed sea level pressure contours around local minima, along with the refinements by Sprenger et al.
(2017) and the extension to multi-center cyclones by Hanley and Caballero (2012). For this study, the algorithm had to be adapted for limited-area domains. Additionally, tracking over time is based on the full extent of the two-dimensional features (see Appendix A) as opposed to only their center points as in the original tracking scheme by Wernli and Schwierz (2006) and Sprenger et al. (2017). As input fields, instead of sea level pressure, we use geopotential ($\Phi$) at 850 hPa for the sake of consistency with the fronts identified at that level. Seasonal feature and track frequency composites are provided in the
supplementary material (Figs. S.S1 and S2)

The $\Phi$ field is first smoothed with a Gaussian filter to eliminate spurious extrema on the high-resolution grid. In order to avoid artifacts, we exclude areas within two grid points ($\sim 24$ km) from the boundaries. Contours are then identified at an interval of $1\,\mathrm{m^2\,s^{-2}}$. Following Wernli and Schwierz (2006) and Sprenger et al. (2017), the outermost enclosing contour around each local minimum is detected by stepping through its enclosing contours until there is no further enclosing contour, or until the next
contour also encloses either a local maximum or a fourth local minimum – the last criterion being a consequence of allowing up to three local minima per cyclone following Hanley and Caballero (2012). Two depth criteria are applied to eliminate spurious minima, whereby the depth of a cyclone feature corresponds to the difference in $\Phi$ between a local minimum and its outermost enclosing contour. First, multi-center cyclone features that are too shallow are split into multiple single- or double-center cyclone features, using the same approach (based on the relative depth of saddle points between minima) and thresholds as
Hanley and Caballero (2012). Second, very shallow cyclone features with a total depth below $1\,\mathrm{m^2\,s^{-2}}$ are discarded.

The approaches by Wernli and Schwierz (2006), Sprenger et al. (2017), and Hanley and Caballero (2012) were developed for global data sets. Limited-area domains introduce an additional complication because it is impossible to determine whether contours that leave the domain are open or closed, or whether they contain additional minima or maxima outside the domain.





It is not obvious how to best deal with such boundary-crossing contours. At one extreme is the assumption that all boundary-
crossing contours are open, which inhibits further growth of cyclone features at the boundaries and thus severely limits the
size of cyclones in the vicinity of the boundaries. The assumption at the other extreme is that all boundary-crossing contours
are closed, which, however, allows for unreasonably large cyclone features in certain situations with a relatively flat pressure
distribution. We opt for a compromise by allowing one in five contours of a feature (20 %) to cross the boundary before halting
further feature growth.

## 2.3 Fronts

The front identification approach is based on Jenkner et al. (2010) and constitutes a multi-step procedure:

- compute fields of frontal strength and velocity;

- based on them, identify cold-frontal and warm-frontal areas as two-dimensional features;

- track these features over time; and

- categorize the resulting front tracks further as either synoptic or local.

Seasonal feature and track frequency composites are provided in the supplementary material (Figs. S1 and S2).

Fronts are characterized by strong horizontal contrasts in low-level temperature and humidity, which qualifies equivalent
potential temperature $\theta_e$ at 850 hPa as a suitable field for their detection (specifically, we use the modulus $|\nabla\theta_e|$ of the $\theta_e$
gradient. Schemm et al. (2018) discuss this choice in detail and provide a historical context. Following the general approach
proposed by Hewson (1998), the front identification method developed by Jenkner et al. (2010) is based on applying the thermal
front parameter (TFP) (Renard and Clarke, 1965) to $\theta_e$ and using the cross-frontal wind component to distinguish between cold,
warm, and quasi-stationary fronts.

Input fields are smoothed with the diffusive filter described by Jenkner et al. (2010) with 160 repetitions. Further noise
reduction is achieved by computing the gradient at each grid point across multiple unit grid distances using offsets of $(i\pm4, j\pm
4)$ instead of $(i\pm1, j\pm1)$.

The frontal areas are derived from a thermal and a wind component:

- The thermal component is based on $|\nabla\theta_e|$ at 850 hPa, from which a mask is derived by applying a minimum threshold.
  The latter varies over the year to account for the strong seasonal cycle of humidity (and therefore of $\theta_e$) leading to
  substantially lower cross-frontal $\theta_e$ gradients in winter than in summer, and thus far fewer winter than summer fronts
  for a given threshold (Rüdisühli, 2018). A threshold value is defined in the middle of each month (Table 1) and linearly
  interpolated to each hour in-between.

- The wind component is based on frontal velocity $v_f$ at 850 hPa:

$$v_f = \boldsymbol{v} \cdot \frac{\nabla\,(\mathrm{TFP})}{|\nabla\,(\mathrm{TFP})|} \qquad (1)$$





where $v$ is the horizontal wind vector and TFP denotes the thermal front parameter, defined as:

$$\text{TFP} = -\nabla \left| \nabla \theta_e \right| \cdot \frac{\nabla \theta_e}{\left| \nabla \theta_e \right|} \tag{2}$$

A mask is derived with $\left| v_f \right| \geq 1 \, \text{m} \, \text{s}^{-1}$.

The frontal areas correspond to the overlap between the thermal and the wind component masks. The sign of $v_f$ determines whether an area is classified as cold-frontal ($v_f \geq 1 \, \text{m} \, \text{s}^{-1}$) or warm-frontal ($v_f \leq -1 \, \text{m} \, \text{s}^{-1}$).

In a next step, the frontal features are tracked over time using the tool described in Appendix A. Cold-frontal and warm-
frontal features are tracked separately. A minimum lifetime criterion of 24 h is applied to discard short-lived fronts. The resulting front tracks are then grouped into synoptic and local fronts based on track properties. Local fronts – largely produced by differential heating along topography and coasts – are generally smaller and more stationary than synoptic fronts. These properties can be expressed by a pair of criteria (on which we have settled after extensive manual testing):

- The *typical feature size* of a track is calculated by first combining, at each time step, the sizes of all features that belong
to the track; and then calculating the median of these total sizes over all time steps. Front tracks are considered local if the *typical feature size* does not exceed $1000 \, \text{km}^2$.

- The *stationarity* of a track is determined as its total footprint area (defined by all grid points that belong to the tracked front at any time) divided by the *typical feature size*. Front tracks are considered local if the *stationarity* does not exceed 6.0.

All tracks fulfilling one or both criteria are considered local fronts, and thus small and/or stationary. All remaining tracks are considered synoptic fronts, and thus both large and non-stationary.

## 2.4 High-Pressure Areas

Precipitation not only occurs near cyclones and fronts, but also in areas of weak synoptic forcing typically characterized by relatively high pressure or by a flat pressure distributions, for example with diurnal summer convection over the continent. We
explicitly identify such high-pressure areas based on geopotential $\Phi$ and its gradient $\nabla \Phi$ at 850 hPa. Seasonal feature frequency composites are provided in the supplementary material (Fig. S1).

The $\Phi$ field is first smoothed with a Gaussian filter. A mask is derived by applying a minimum threshold that varies over the year to account for the seasonal cycle in $\Phi$. Analogous to the seasonally varying frontal threshold, the $\Phi$ threshold values are defined in the middle of each month (Table 2) and linearly interpolated to each hour in-between. Then, $\nabla \Phi$ is computed, and
the resulting field is smoothed again. A second mask is derived by applying a constant maximum threshold of $0.02 \, \text{m} \, \text{s}^{-2}$ to $\nabla \Phi$. The high-pressure area corresponds to the overlap area of the $\Phi$ and $\nabla \Phi$ masks. All threshold values have been determined subjectively based on thorough manual testing.





## 2.5 Front-Cyclone-Relative Components

In order to associate precipitation to fronts and cyclones, we decompose the domain at each time step into seven so-called
front-cyclone-relative components, as illustrated in Fig. 2. They are mutually exclusive, with each grid point assigned to the
first component for which it fulfills the criteria, and defined in the following order:

1. The **high-pressure** component comprises all grid points within a high-pressure area mask (regardless of the presence
   of fronts or cyclones). Its purpose is to capture precipitation in areas of weak synoptic forcing such as diurnal summer
   convection over the continent. Applying this criterion first prevents spurious front features that frequently occur in the
   Mediterranean in summer from capturing diurnal summer convection precipitation as far-frontal.

2. The **cyclonic** component comprises all remaining grid points within a cyclone mask, regardless of the presence of fronts.
   Its purpose is to capture precipitation produced near the center of cyclones.

3. The **cold-frontal** component comprises all remaining grid points within 300 km of a cold-frontal feature, but farther
   than 300 km from a warm-frontal feature. Its purpose is to capture all precipitation produced close to cold fronts but in
   relative isolation from warm fronts and cyclone centers.

4. The **warm-frontal** component is analogous to the cold-frontal, but for warm fronts.

5. The **collocated** component comprises all remaining grid points within 300 km of both a cold-frontal and a warm-frontal
   feature. Its purpose is to capture precipitation simultaneously influenced by cold and warm fronts away from cyclone
   centers, for instance, in areas of frontal fracture or frontal occlusion. In addition, it also occasionally captures strong
   warm conveyor belts, the eastern boundaries of which can be associated with a band of very high $\theta_e$ that is identified as
   a warm front just ahead of the cold front.

6. The **far-frontal** component comprises all remaining grid points within 300–600 km of a front of either type. No distinc-
   tion is made between the front type in order to keep the number of groups reasonably small. Its purpose is to capture
   precipitation more remotely related to fronts.

7. The **residual** component comprises all remaining grid points. Its purpose is to capture precipitation that our approach
   cannot attribute to a specific weather system. Under the assumption that the other six components capture the major
   sources of precipitation, we expect the residual contributions to be comparatively small.

The thresholds that define the near-frontal (300 km) and far-frontal (600 km) components have been chosen subjectively based
on our best judgment while studying a range of cases.

## 275   3   Case Studies

In order to illustrate our approach, we present two case studies of a summer and a winter cyclone.



### 3.1 Summer Cyclone Uriah

In late June 2007, cyclone Uriah (FU Berlin, 2007b) moved across the British Isles and the North Sea, accompanied by a strong cold front and a weak warm front.

#### 3.1.1 Development

At 06 UTC 25 June 2007 (Fig. 3 a), the cold front is part of a baroclinic zone that extends from northeastern France southwestward to Gibraltar. The main precipitation areas are located just north of the cyclone center, as well as along and ahead of the cold front over France and Germany. East of the cyclone center, a weaker warm-frontal zone extends into Eastern Europe, but it is not yet recognized as a front feature. Northwest of the cyclone center, cold and dry air is advected southward. The southern

boundary of this cold zone constitutes a weakly precipitating cold front approaching the British Isles.

At 15 UTC 25 June 2007 (Fig. 3 b), the main cold front is gaining strength while moving over France and Germany. Its northern end has started to wrap around the cyclone center and produces substantial precipitation, while its southern end has reached the Alps, producing strong precipitation along the northern flank of the Alps. Behind the cold front, over France, many isolated cells produce fragmented precipitation of weak to moderate intensity. The warm front east of the cyclone, now detected

as a feature, is much weaker than the cold front and produces no precipitation, except close to the cold front, where occlusion may have commenced. The baroclinic zone southwest of the cyclone center has been fragmented while moving over Iberia and France. The minor cold front to the northwest of the cyclone center has reached Scotland and Ireland while falling dry.

At 06 UTC 26 June 2007 (Fig. 3 c), the main cold front has moved from Germany over Eastern Europe and southern Scandinavia. It is mostly oriented northwest-southeastward, except for its northern end, which is bent around the cyclone

center. Precipitation is still substantial along most of the front. The precipitation band along its bent-back portion wraps almost completely around the cyclone center, much farther than the respective front feature, which suggests that not the whole front has been detected as a feature by our algorithm. The southern end of the cold front has been held back along the Alps, but orographic precipitation has largely stopped. In the cold sector behind the front, over France and Germany, fragmented postfrontal precipitation is still prevalent. The warm front is now detected as a pair of thin warm-frontal features. Along most

of its length, the warm front has been caught up by the cold front, suggesting occlusion. The baroclinic zone consisting of many small frontal fragments has crossed the Spanish and French coast into the Mediterranean. The minor cold front over Great Britain at the boundary of the cold zone has stopped precipitating and is now followed by a pair of likewise dry warm fronts along the western border of the cold zone.

#### 3.1.2 Precipitation Attribution

Figure 4 shows the attribution to fronts and cyclones of the accumulated precipitation during the four days when cyclone Uriah affected Europe. We have characterized Uriah as a slow-moving cyclone accompanied by a pronounced cold front and no discernible warm front. The precipitation attribution is entirely consistent with that. Most precipitation accumulated in a ring-shaped area centered on the Danish Straits, with maxima over the North Sea and southern Sweden (Fig. 4 a). The precipitation





area extends over the British Isles and France to the west and southwest, and southward to the Alps; along the northern flank,

precipitation amounts are locally enhanced. Southern Europe and the Mediterranean are entirely dry. Most precipitation is classified as either cyclonic (Fig. 4 f), mainly over the North Sea, southern Scandinavia, and the Baltic Sea; or cold-frontal, mainly over Germany and Poland near the Baltic coast and extending southwestward to the Alps (Fig. 4 a). While there is also some far-frontal and residual precipitation (Fig. 4 g, h), there is essentially no warm-frontal, collocated, or high-pressure precipitation (Fig. 4 c, d, e).

### 3.2 Winter Cyclone Lancelot

Winter storm Lancelot (FU Berlin, 2007a) affected Europe during 19–21 January 2007 in the wake of well-known winter storm Kyrill (see Leutwyler et al. (2015) for an animation based on the same simulation).

#### 3.2.1 Development

At 00 UTC 20 January 2007 (Fig. 5 a) the cyclone center approaches Ireland, accompanied by a warm front extending south-

eastward into the North Sea and Central Europe, and a cold front extending southwestward across the British Isles into the North Atlantic. A large area of precipitation associated with the warm front extends over the North Sea to the rear of the cyclone center. A smaller band of precipitation accompanies the cold front, separated from the warm-frontal precipitation area by a dry gap region.

  At 12 UTC 20 January 2007 (Fig. 5 b), the cyclone center has almost completely crossed the North Sea and is approaching

the southern tip of Norway. The cold front has been moving away from the cyclone center toward the southeast. It is oriented at a right angle to the warm front, forming a frontal T-bone typical of Shapiro-Keyser-type cyclones (Shapiro and Keyser, 1990). The dry gap region between the fronts has disappeared. Along the cold front, oval-shaped precipitation cores are discernible, which are oriented at a slight clockwise angle relative to the front and separated by gap regions, reminiscent of a narrow cold-frontal rainband. In the cold sector behind the cyclone, there is widespread patchy precipitation, some of it associated with a

relatively shallow cyclone near the British Isles.

  At 00 UTC 21 January 2007 (Fig. 5 c), the cyclone center resides over the southern Scandinavian Peninsula. The warm front has moved across the southern Baltic Sea while still producing precipitation over an extended area. The cold front, by now far away from the cyclone center, has moved over continental Europe, extending from the Atlantic near the northern tip of Iberia across France and Germany into Eastern Europe. It is oriented roughly parallel to the Alpine crest, while steadily approaching

it. The eastern part of the cold front over Germany and Eastern Europe has started to disintegrate.

#### 3.2.2 Precipitation Attribution

Figure 6 shows the attribution to fronts and cyclones of the accumulated precipitation during the three days when cyclone Lancelot affected Europe. It moved faster and more zonally than cyclone Uriah and was accompanied by a large warm front and a long cold front, all of which is reflected in the precipitation contributions. Accumulated precipitation is distributed across





most of the northern half of the domain, with a pronounced local maximum along the northern flank of the Alps (Fig. 6 a).
In addition, local maxima occur over and west of Scotland, over southern Norway, and to a lesser degree over Denmark and
along the Baltic coast. Like during the passage of cyclone Uriah, the Mediterranean is dry. Most components contribute some
precipitation except, for high-pressure areas (Fig. 6 e). Much precipitation is classified as frontal, with cold-frontal precipitation
mainly north of the Alps (Fig. 6 b), warm-frontal precipitation covering an elongated region extending from the North Sea

across Denmark into Poland (Fig. 6 c), and large amounts of collocated precipitation organized in two distinct band-like regions
farther south and north (Fig. 6 d). The precipitation maximum along the Alps is primarily collocated – however, that southern
region of collocated precipitation largely predates the passage of Lancelot and is at least partially caused by remnants of Kyrill,
the cyclone system immediately preceding Lancelot, as is evident in Fig. 5 a. Also attributable is some cyclonic precipitation
over southern Scandinavia and the Baltic (Fig. 6 f), along with some scattered far-frontal precipitation (Fig. 6 g). Residual

precipitation is largely restricted to the northern part of the British Isles and the adjacent North Atlantic (Fig. 6 h). As Fig. 5 b, c
indicate, post-frontal precipitation was largely responsible for the residual, partly organized in secondary frontal and cyclonic
structures not identified as synoptic features.

These case studies illustrate that our method is able to attribute precipitation to cyclones and fronts meaningfully and to
capture the large case-to-case variability of the various contributions.

## 4  Climatology


In this section, the nine-year (2000-2008) climatology of precipitation and its link to the features in Fig. 2 are discussed. First,
we consider the total precipitation in Sec. 4.1, whereby the annual and seasonal climatologies are discussed separately. Then,
we focus on heavy precipitation in Sec. 4.2.

### 4.1  Total Precipitation

The main results of the total precipitation attribution are shown in Fig. 7 for absolute annual-mean amounts, Fig. 8 for absolute
seasonal-mean amounts, and Fig. 9 for relative seasonal-mean contributions. In the annual mean, the total precipitation amounts
are generally larger in the northern part of the domain than in the Mediterranean. The largest amounts, however, occur over high
topography, especially the Alps, the Dinaric Alps, the Norwegian Alps, the Scottish Highlands, and the Pyrenees. In the North
Atlantic, the precipitation amounts decrease from north/northwest toward south/southeast. With respect to the front-cyclone-

relative contributions, several interesting features are discernible: (i) cold-frontal precipitation amounts are largest over the Alps
and still large to the north/northwest thereof, but rather small in the Mediterranean and the Baltic Sea; (ii) large warm-frontal
amounts are found over the North Atlantic and (to a lesser degree) over Central Europe, but they are also almost absent over
the Mediterranean; (iii) cyclonic precipitation is rather uniformly distributed across the domain with peak values in the North
Atlantic, the British Isles, Northern Scandinavia, and the Mediterranean, which makes it the only component that substantially

contributes to the Mediterranean precipitation; (iv) the high-pressure precipitation amounts are large along a continental band
extending from the Pyrenees to the Alps and the Dinaric Alps, with a further band extending along the Apennines; and (v) the





residual precipitation (i.e., the amounts that cannot be attributed to any front-cyclone-relative component), are rather evenly distributed in the domain, with enhanced values only discernible over the Alps and the Norwegian Alps.

The discussion so far has ignored the fact that there are significant seasonal variations (Fig. 8). In winter, the total precipita-
tion is shifted from the continental regions to the North Atlantic. In spring, the distribution is close to the annual mean, except for slightly below-average amounts in the North Atlantic, the Baltic, and the Mediterranean Sea, and slightly more precipitation over the Alps, the Pyrenees, and the Dinaric Alps. In summer, the spatial distribution across the domain is the least uniform among all seasons: The Mediterranean Sea and the Iberian peninsula are almost completely dry; most of continental Europe receives more precipitation than on average; and the contrast between the large precipitation amounts over the Alps and the
dryer surrounding areas is biggest by far. Furthermore, during summer, no peak amounts are discernible over the Pyrenees and the Dinaric Alps, quite in contrast to spring and fall. Finally, the precipitation in fall is similarly distributed as in the annual mean, except for larger precipitation amounts in the North Atlantic relative to continental Europe. Peaks amounts in fall occur over the Alps, the Norwegian Alps, the Pyrenees, the Dinaric Alps, and the Scottish Highlands, as they do in the annual mean.

In the same way as the amounts and geographical distribution of precipitation exhibit a distinct seasonal dependence, sea-
sonal variations can also be expected for the front-cyclone-relative components. Physically, this is, of course, based on the seasonal cycle of the considered weather features (cold and warm fronts, cyclones, high-pressure areas). For instance, it is well known that lee cyclones form preferentially during spring and fall in the Gulf of Genoa (e.g., Campins et al., 2011), or that North Atlantic cyclones with their attendant cold and warm fronts affect continental Europe more often in winter than in summer (e.g., Hénin et al., 2019). Seasonal variations in the relative front-cyclone precipitation amounts must, therefore, be
expected and interpreted with respect to the corresponding shifts in the weather features. In the supplementary material, we provide seasonal climatologies of fronts, cyclones, and high-pressure areas (Figs. S1 and S2) along with the occurrence and wet-hour frequencies of the front-cyclone-relative components (Figs. S3–S6). Here, we restrict the discussion to a few select seasonal effects on the relative precipitation amounts: (i) The cold-frontal precipitation is more uniformly distributed across the domain in winter and fall than in spring and summer, whereby in summer, cold-frontal precipitation is mostly restricted
to the continent, specifically Western, Eastern, and Northern Europe; (ii) warm-frontal winter precipitation is similarly dis-
tributed as the annual mean – peak values over the North Atlantic and the British Isles, and somewhat smaller values over Central Europe – whereas summer warm-frontal precipitation is nearly non-existent over the continent; (iii) cyclonic winter precipitation amounts are below the annual average over continental Europe and the North Atlantic, but above-average in the Mediterranean, especially in the Adriatic and Tyrrhenian Seas and over the Apennines, in contrast to summer with nearly no
cyclonic precipitation over the Mediterranean Sea; and (iv) high-pressure precipitation dominates in summer over much of Western and Southeastern Europe, whereas this component is completely missing during winter and only weakly discernible in spring and fall. This short list, of course, can only provide a glimpse on the many local seasonal effects. Furthermore, as mentioned before, we did not show and describe the seasonality of the collocated and far-frontal components, which, however, can be found in the supplementary material (Figs. S3–S6).

Instead of analyzing in greater detail the *absolute* precipitation amounts and how they can be attributed to the front-cyclone-
relative components, we now consider the *relative* contributions by addressing the questions what percentage of the total





precipitation can be attributed to a front-cyclone system, and what percentage is attributable to either the high-pressure or residual components. The results are shown in Fig. 9, split according to season and for the components: frontal, i.e., cold- or warm-frontal; cyclonic and far-frontal; high-pressure; and residual. There are several signals discernible that are noteworthy.

During winter, a substantial percentage ($> 70\%$) of the total precipitation can be attributed to fronts over the North Atlantic, Central Europe, and (to a lesser degree) the Mediterranean. Cyclonic and far-frontal percentages are largest in the Mediterranean, particularly in spring (regionally up to 50 %). High-pressure percentages are negligible except for summer when the contribution over the Iberian peninsula, mid- to southern Italy, and Sardinia/Corsica becomes larger than 70 %. Of course, part of the total precipitation cannot be attributed to any of the components. The relative residual contributions are rather uniform,

both in time and in space. In spring, they reach about 25 %. Especially in winter and fall, the residual percentages over Central Europe and the North Atlantic, including the British Isles, are still smaller, about 10 %.

## 4.2 Heavy Precipitation

After the discussion of total precipitation in the previous section, we now shift our focus to heavy precipitation. It is defined as the amount of precipitation exceeding a local (i.e., grid-point-specific) threshold of hourly precipitation intensity, which

corresponds to the $99.9^{\text{th}}$ all-hour percentile (i.e., including dry hours, as recommended by Schär et al., 2016), corresponding to a return period of about 1.4 months. Separate thresholds are computed for annual and seasonal analyses, respectively.

The spatial distribution of the annual-mean heavy precipitation (Fig. 10 a) differs from total precipitation (Fig. 7) in that the former preferentially occurs over land; and that heavy precipitation amounts in the Mediterranean are similar to those over continental Europe and larger than those in the North Atlantic. While total precipitation exhibits the strongest spatial gradients

from low to high topography, especially in the Alpine region, heavy precipitation shows a more pronounced land-sea-contrast, especially between the North Atlantic and continental Europe. Local maxima in heavy precipitation amounts occur over high topography along the northern Mediterranean, specifically over the Alps, the Pyrenees, the Dinaric Alps along the Balkan coast, and the Apennines.

The front-cyclone-relative components of annual-mean heavy precipitation can be sorted into two groups: (i) cold-frontal,

high-pressure, cyclonic, and residual precipitation (Fig. 10 b, e, f, h), which each contribute substantial amounts of heavy precipitation in specific areas; and (ii) warm-frontal, collocated, and far-frontal heavy precipitation contributions (Fig. 10 c, d, g), which are substantially smaller and will therefore not be discussed any further. Some specific attribution results with respect to the first group are: (i) Cold-frontal heavy precipitation (Fig. 10 b) is large over and around the Alps, as well as along the Balkan and the northwestern Iberian coasts; (ii) high-pressure heavy precipitation (Fig. 10 e) is restricted to continental areas,

both Europe and North Africa, and contributes by far the largest share to heavy precipitation over land; (iii) cyclonic heavy precipitation (Fig. 10 f) resembles total cyclonic precipitation in its relatively even spatial distribution and only weak local enhancement over high topography, while contributing almost all heavy precipitation over the Mediterranean Sea and to a lesser degree in the North Atlantic and the North Sea; and (iv) residual heavy precipitation amounts (Fig. 10 h) tend to be larger over land than over sea and increase toward Eastern Europe, albeit in contrast to total precipitation without any local enhancement

over high topography.





Like total precipitation, heavy precipitation exhibits seasonal variations in both geographical distribution and front-cyclone-relative attribution. The clear separation into the two above-mentioned groups in the annual mean disappears, which reflects the fact that different mechanisms are responsible for heavy precipitation in different seasons – which is expected given the seasonality of the considered weather features (see supplementary material, Figs. S1 and S2). Heavy winter precipitation (Fig. 11 a)

is more prevalent over sea than over land, as opposed to the annual mean, with the largest amounts over the Mediterranean – and especially the Ionian – Sea, as well as along the Iberian west coast. In spring, heavy precipitation (Fig. 11 a) exhibits a pronounced land-sea-contrast with large amounts distributed evenly across continental Europe and local maxima over the Alps and the Tunesian Atlas. Compared with winter, this corresponds to pronounced a north- and landward shift of heavy precipitation in the southern part of the domain. No season experiences more heavy precipitation than summer (Fig. 11 a)

when the northward shift since winter reaches its peak. Heavy precipitation amounts are large over all of continental Europe except Iberia, with peaks over the Alps, and moderate further north over the British Isles, the Baltic, and the North Atlantic. Meanwhile, the Mediterranean Sea and southern Iberia are almost dry. The onset of fall is accompanied by a southward shift of heavy precipitation from continental Europe to the Mediterranean (Fig. 11 a). The spatial distribution is almost mirrored, with most heavy precipitation in the previously dry Mediterranean and Iberia while the land-sea-contrast along the rest of the

North Atlantic coast completely disappears. Italy and the Balkan coast are the only extended regions where heavy precipitation is prevalent in both summer and fall. By far the largest heavy precipitation amounts occur along the coasts of France and Spain from the Gulf of Lion to the Balearic Sea, along with secondary hot spots int the Tyrrhenian and Ionian Seas.

Heavy precipitation is attributable to different processes from season to season (Fig. 11), just as we have already shown for total precipitation (Fig. 8): (i) The main areas of heavy winter precipitation in the Ionian Sea and along the western Iberian

coast originate primarily from, respectively, cyclones (Fig. 11 d), and cold and warm fronts (especially the former; Fig. 11 b, c); (ii) similarly, the cyclonic component (Fig. 11 d) is the primary source of heavy precipitation in the Mediterranean in the other seasons, especially in fall, and over Northern Europe and the North Atlantic in summer; (iii) the widespread occurrence of heavy summer precipitation over the continent almost entirely coincides with high-pressure areas (Fig. 11 e), which on the other hand are completely irrelevant in winter; and (iv) while cold fronts (Fig. 11 b) steadily contribute heavy precipitation

over the continent from spring through fall, with peak contributions along the northwestern Mediterranean coast (Gulf of Lion, Gulf of Genoa) in fall, warm fronts (Fig. 11 c) are mostly irrelevant for heavy precipitation.

## 5   Conclusions

Hourly fields from a kilometer-scale regional climate simulation for present-day climate conditions over Europe, covering the nine-year period 2000–2008, have been used to perform a detailed climatological attribution of total and heavy precipitation to

a set of synoptic weather systems: cyclones, cold and warm fronts, high-pressure areas (capturing diurnal summer convection), and derived categories (regions with collocated cold and warm fronts and far-frontal regions). To the best of our knowledge, this is so far the most detailed synoptic feature attribution exercise for European precipitation, which led to important findings related to both methodological and meteorological aspects. First, the attribution has been applied to two storms passing over





Europe: the summer cyclone Uriah (24–26 June 2007), and the winter cyclone Lancelot (19–21 January 2007). Based on
these two case studies, and further refined in the 2000-2008 climatological analysis, the methodological key aspects can be
summarized as follows:

- Although fairly established algorithms existed for automatically identifying cyclones and fronts in comparatively coarse
  reanalysis and global climate simulation data, their application required great efforts in testing and adjusting for use with
  kilometer-scale simulation output (e.g., by increasing spatial smoothing and by introducing additional criteria). These
  efforts can hardly be automated, and the finally used thresholds are not universal, i.e., they would need further adjustment
  if considering a different region, climate model, or resolution. The final setup of our algorithms should not be regarded
  as perfect, but rather pragmatically as one out of potentially several meaningful options.

- A large model domain is required in order to meaningfully identify frontal cyclones, in particular in the North Atlantic
  storm track region. Although, compared with previous kilometer-scale climate simulations, our simulation was per-
  formed on a huge domain, it was essential to perform the identification of cyclones and fronts on the even larger domain
  of the driving coarser model. Only with this spatial extension, the robust identification of North Atlantic cyclones and
  their sometimes elongated trailing fronts approaching Europe became possible.

- A particular challenge related to the frontal identification is the choice of the equivalent potential temperature gradient
  threshold. If a constant threshold is used, a spuriously high number of fronts appear in summer, while a substantial
  number of fronts are missed in winter. We therefore introduced a seasonally varying gradient threshold, which led
  to a fairly constant number of identified fronts throughout the year. However, this clearly emphasizes the degree of
  subjectivity associated with the identification of fronts, which directly affects the attribution of precipitation to those
  fronts.

The meteorological results of the precipitation attribution show that different components are important in different ge-
ographical regions and in different seasons. When considering precipitation over the entire year, the most relevant weather
systems are cold fronts near the Alps, warm fronts and cyclone centers in the North Atlantic and Western Europe, and cy-
clones in the Mediterranean, in particular near Italy and the Balkans. A substantial residual exists (about 20–30 %), indicating
that our weather system categories do not encompass all precipitation-producing flow situations and that the attribution to the
target systems is not perfect. Strong local enhancement occurs over high topography compared to the sourrounding flat areas,
which is especially pronounced over the Alps and for cold-frontal precipitation. From a seasonal perspective, (i) cold fronts are
important contributors in all seasons (especially over the continent), while warm fronts primarily contribute in winter and fall
(especially over the North Atlantic); (ii) the largest cyclonic contributions shift from the Mediterranean in winter to Northern
Europe in summer; and (iii) high-pressure precipitation is confined to summer over the continent, with pronounced local en-
hancement over the Alps. Focusing only on heavy-precipitation events reveals substantial differences to total precipitation: (i)
Rather than over high-topography, heavy precipitation is particularly enhanced over land compared to sea; (ii) cold fronts also
contribute substantially to heavy precipitation, whereas the relevance of warm fronts diminishes; (iii) cyclones are particularly





important for heavy precipitation over the ocean; and (iv) the summertime high-pressure systems further gain in significance, in particular for continental summer convection.

The results can be summarized concisely for several distinct geographical regions. In particular, we focus on (i) the British
Isles, (ii) Western Europe (excluding the Alps), (iii) the Alps, (iv) Southeastern Europe (comprising Italy, Corse, and the Balkan coast), (v) the Iberian Peninsula, and (vi) the Mediterranean Sea. The mean precipitation amounts over the whole domain and each region for all front-cyclone-relative components in each season are shown in Fig. 12. Of course, this selection of geographical regions is not exhaustive, and could easily be extended to other regions based on the distribution maps in this study (Figs. 7 to 11) and the supplementary material (Figs. S2–S6).

– **British Isles:** Cyclonic and frontal precipitation are important throughout the year, but there is also a clear seasonal cycle: The cold-frontal contributions are larger in winter and fall than in spring and summer; warm-frontal contributions – which are larger than for any other region – exhibit a similar but more pronounced seasonal cycle as cold-frontal; and while the cyclonic contributions are relatively weak in winter, they are substantial in spring, fall and particularly summer. High-pressure precipitation plays no role for the British Isles. For heavy precipitation, the importance of warm fronts
diminishes while that of cyclones further increases, and while cyclones experience a more pronounced seasonal cycle with a shift from winter to summer, the seasonality of cold fronts markedly decreases.

– **Western Europe:** Cold-frontal precipitation remains important and uniform in its amplitude in Western Europe throughout the year. By contrast, half the annual warm-frontal precipitation is contributed in winter and almost none in summer. The relevance of cyclones, by contrast, is lowest in winter and peaks in spring. High-pressure precipitation only sub-
stantially contributes in summer, but then more than any other component. With respect to heavy precipitation, cold fronts remain the main contributors overall, but no single-season contribution over Western Europe compares to that of high-pressure areas in summer, which equals or exceeds the annual contributions of all components except cold fronts and cyclones.

– **Alps:** The Alps stand out in many maps as a region with considerably enhanced precipitation amounts. In all seasons,
cold-frontal precipitation contributes substantially to the total precipitation amounts, whereby this signal is particularly strong during spring. Warm-frontal precipitation, on the other hand, is substantially reduced compared to cold-frontal and mostly restricted to fall and winter. Cyclonic and high-pressure precipitation are of equally high overall importance, but while the former exhibits a comparatively weak seasonal cycle, high-pressure precipitation primarily occurs in summer. The residual is notably large over the Alps, especially in spring and summer. This changes in the heavy-precipitation
limit, though, where summer high-pressure precipitation gains even more relevance, followed in total annual amounts by cold-frontal and cyclonic precipitation.

– **Southeastern Europe:** Similarly to the British Isles, cyclonic precipitation is of great importance to precipitation in Southeastern Europe, but warm-frontal precipitation is not. While the cold seasons are markedly influenced by cold fronts and cyclones, high-pressure systems are more important in summer – although not nearly as dominant as over





540          the Alps. Heavy precipitation exhibits a similar attribution profile, except for large amounts of summer high-pressure precipitation, as observed in many regions.

- **Iberian Peninsula:** Summers are very dry, with hardly any precipitation except relatively small amounts of high-pressure precipitation. The other seasons are strongly influenced by cyclones (especially spring) and cold fronts (especially fall) along with some warm-frontal influence. The fraction of unattributable precipitation is large compared with other regions, especially in spring. Heavy precipitation exhibits a very similar attribution profile except for larger summer high-pressure contributions.

- **Mediterranean Sea:** Cyclonic contribution dominate in all seasons, although in summer, the Mediterranean receives almost no precipitation. Cold and warm fronts together contribute about the same total annual amount of precipitation as cyclones, to which cold fronts contribute about twice as much as warm fronts. The cyclonic dominance is even more pronounced for heavy precipitation, especially in fall, when also the relative cold-frontal contributions increase compared to total precipitation. High-pressure contributions increase in summer and fall for heavy precipitation. While all other regions experience more high-pressure precipitation in summer than in fall, the opposite is true in the Mediterranean Sea; this holds for both total and heavy precipitation.

Many of these results are plausible in the sense that they are consistent with meteorological expectations. We think that the particular value of this study are its objective approach, the quantitative results, and the high-resolution maps (Figs. 7 to 11), which allow one to discover many interesting small-scale characteristics of European precipitation. It is interesting that this approach confirms the strongly opposing character of winter and summer precipitation, the former being very strongly associated with cyclones and fronts, and the latter predominantly with high-pressure systems.

There are different aspects that could be studied in forthcoming analyses. For instance, the results presented in this study show how the precipitation can be attributed to the front-cyclone-relative components under present-day climate conditions. It is, however, an open question whether the attribution to the components will be the same in the future climate. First steps to apply our approach to future climate simulations have been taken, and the results will be presented in a forthcoming publication. As an additional refinement, the frontal precipitation may be split into pre- or post-frontal components or a component at the exact location of the front. Such front-relative precipitation profiles would be rather interesting and further refine our understanding of how precipitation is induced by and thus attributable to cyclone-frontal passages. Preliminary results in this direction look promising (Rüdisühli, 2018).





*Code and data availability.* The data and analysis tools used in this study are available upon request.

570

## Appendix A: Identification and Tracking Algorithm

Weather systems are explicitly identified as two-dimensional features comprised of adjacent grid points (including diagonal neighbors) and with characteristic properties such as size and center position. Tracking these features over time enables further characterization based on their time evolution, for instance by applying lifetime or stationarity criteria. Here, we provide a concise summary of our approach – for more details, the reader is referred to Rüdisühli (2018)[1].

The feature tracking algorithm is designed for data with high resolution in space and time. Whether a feature at one time step (the parent) corresponds to one or more features at the next or previous time step (the children), depends on whether they exhibit sufficient overlap and similar total size. (This matching is done symmetrically both forward and backward in time, so the child features may well temporally precede their parent feature.) Based on these metrics, a tracking probability is computed and used to determine the features that correspond to each other. A connection between a parent and its child features constitutes a tracking event. Its type depends on the number of children and the temporal direction of the connection: continuation (one child), merging/splitting (multiple children, backward/forward), genesis/lysis (no children, forward/backward). The resulting feature tracks can contain an arbitrary number of merging and splitting events, and they are therefore in general not linear, but branched. This also implies that at any given time step, multiple features may belong to separate branches of the same track. The duration of a track is defined as the time difference between its earliest and the latest features, regardless of how the respective branches are connected in-between.

*Author contributions.* SR designed this study together with MS and HW. SR developed the analysis tools and produced the results. DL and CS contributed the output of the high-resolution simulation. SR did most of the writing, and all authors contributed to the discussion of the results and the final manuscript.

*Competing interests.* The authors declare that they have no conflict of interest.

---

[1]Note that in Rüdisühli (2018), additional algorithmic components such as feature and track splitting or topography filters were described and applied to cyclones and fronts. Unless explicitly mentioned, they have not been applied in the present study.



*Acknowledgements.* The Swiss National Science Foundation supported this work under Sinergia Grant CRSII2_154486/1 crCLIM. Compute resources for the decade-long climate simulation were awarded through the Partnership for Advanced Computing in Europe (PRACE) on Piz Daint at the Swiss National Supercomputing Center (CSCS). Furthermore, we acknowledge Nicolas Piaget (formerly ETH Zurich) for technical support during the early stages of this work; Nikolina Ban (University of Innsbruck) for helpful discussions and comments; and Olivia Romppainen-Martius (University of Bern) for helpful feedback on an earlier version of this study.





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



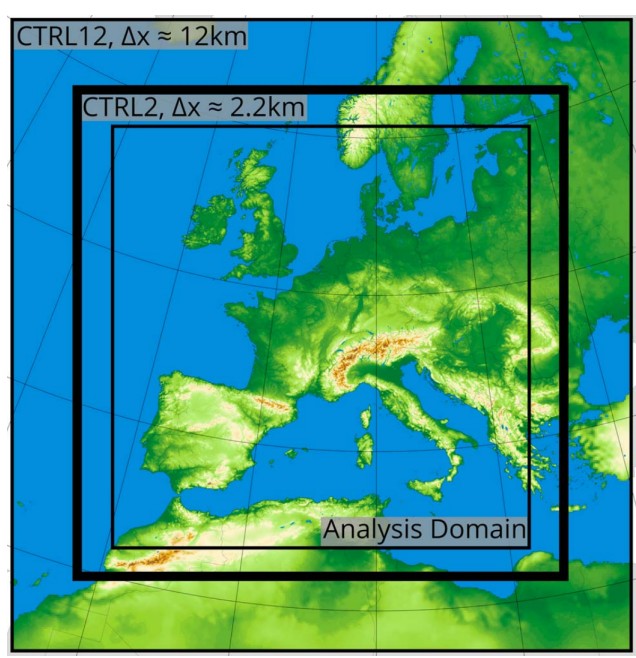

**Figure 1.** COSMO simulation domains and model topography. The outermost black box denotes the domain of the convection-parameterizing simulation with a grid spacing of 12 km and the bold box the domain of the convection-resolving simulation with 2.2 km grid spacing. The innermost thin box indicates the subdomain used in the analysis. (Figure and caption from Leutwyler et al., 2017)





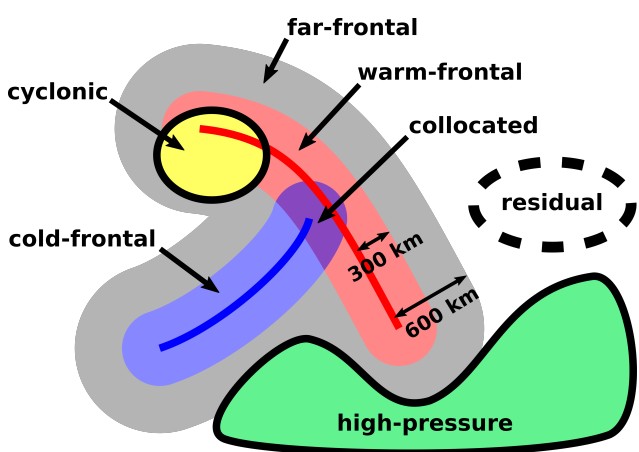

**Figure 2.** Schematic depiction of the seven front-cyclone-relative components high-pressure, cyclonic, cold-frontal, warm-frontal, collocated, far-frontal, and residual, as defined in Sec. 2.5.



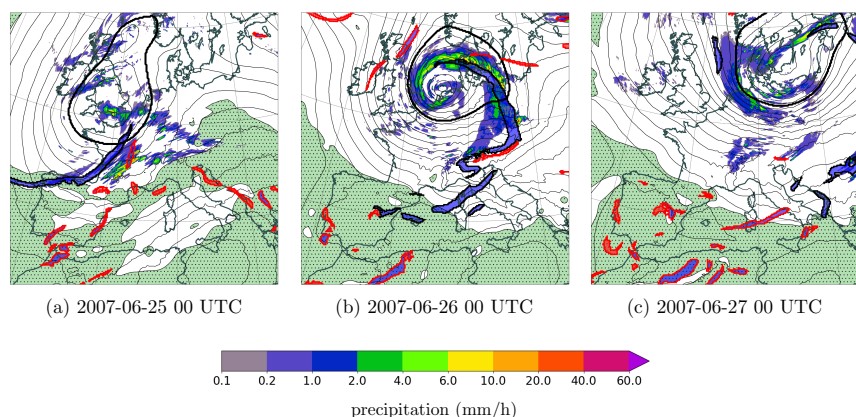

(a) 2007-06-25 00 UTC     (b) 2007-06-26 00 UTC     (c) 2007-06-27 00 UTC

precipitation (mm/h)

**Figure 3.** Development of cyclone Uriah in June 2007. The thin black contours indicate geopotential at 850 hPa; the colored shading the surface precipitation; the filled bold contours the outlines of front features, with black/red outlines for synoptic/local fronts, and blue/red filling for cold/warm fronts; the unfilled bold contours the outlines of cyclone features; and the green stippling high-pressure features.

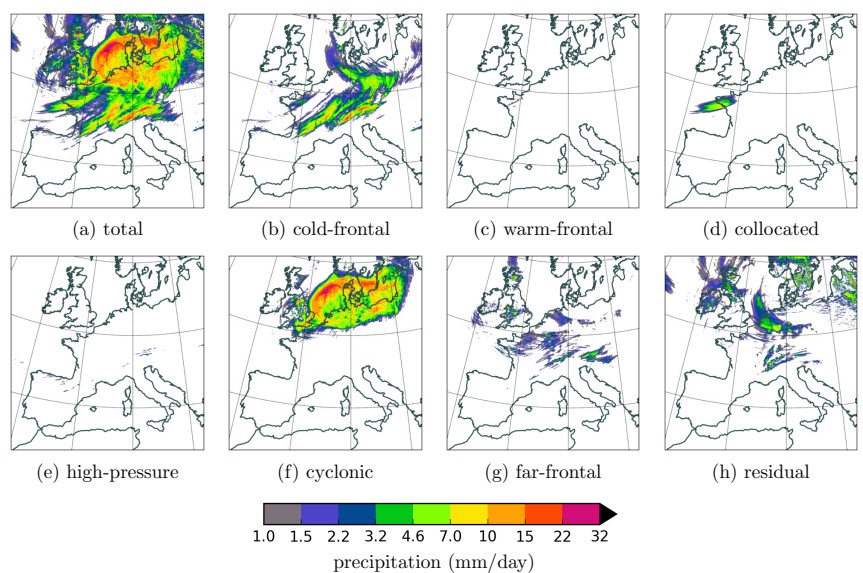

**Figure 4.** Front-cyclone-relative precipitation contributions to cyclone Uriah during the period 24–27 June 2007.



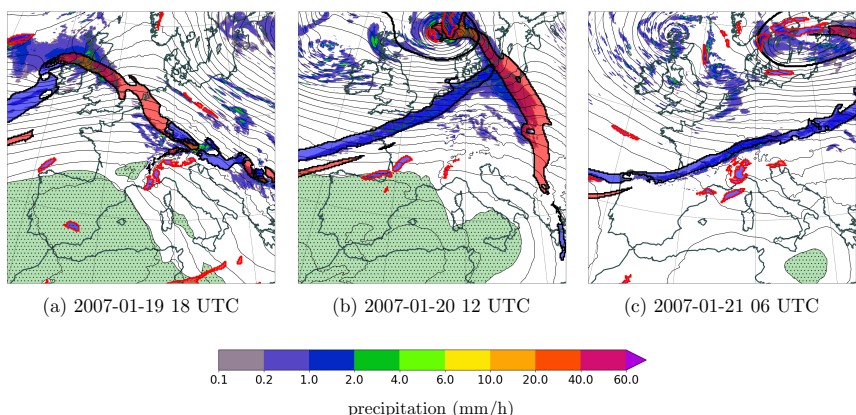

**Figure 5.** As Fig. 3, but for cyclone Lancelot in January 2007.

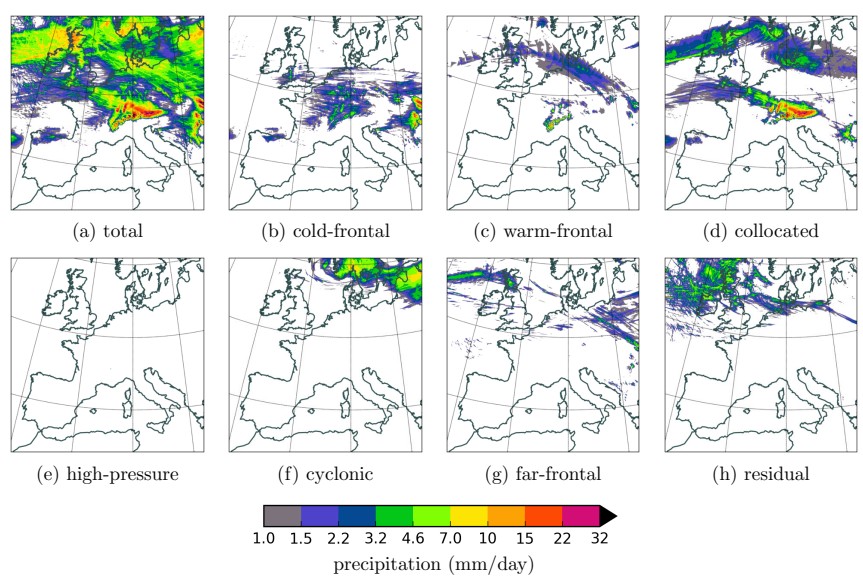

**Figure 6.** As Fig. 4, but for cyclone Lancelot during the period 19–21 January 2007.



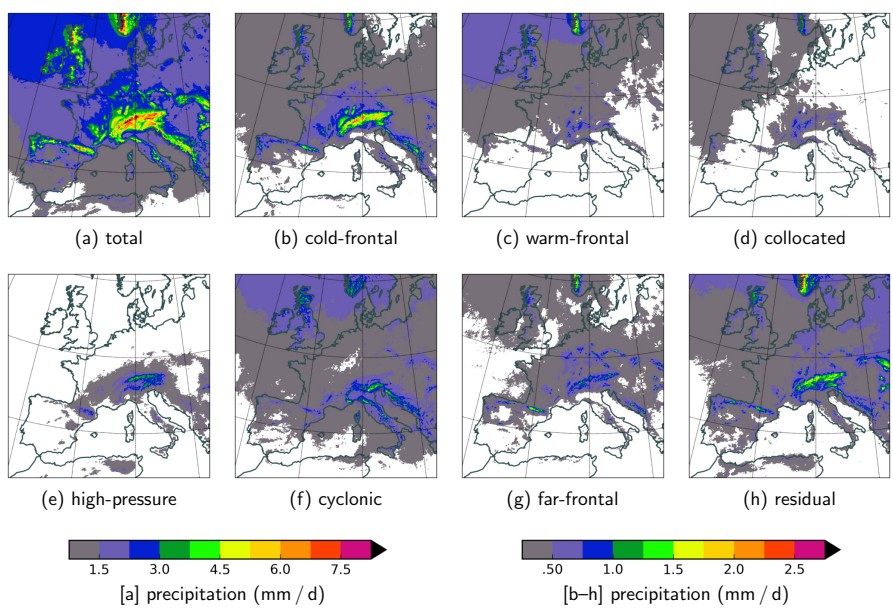

**Figure 7.** Mean daily precipitation during the nine-year period 2000–2008 (a) overall and (b–h) separated into seven front-cyclone-relative contributions.

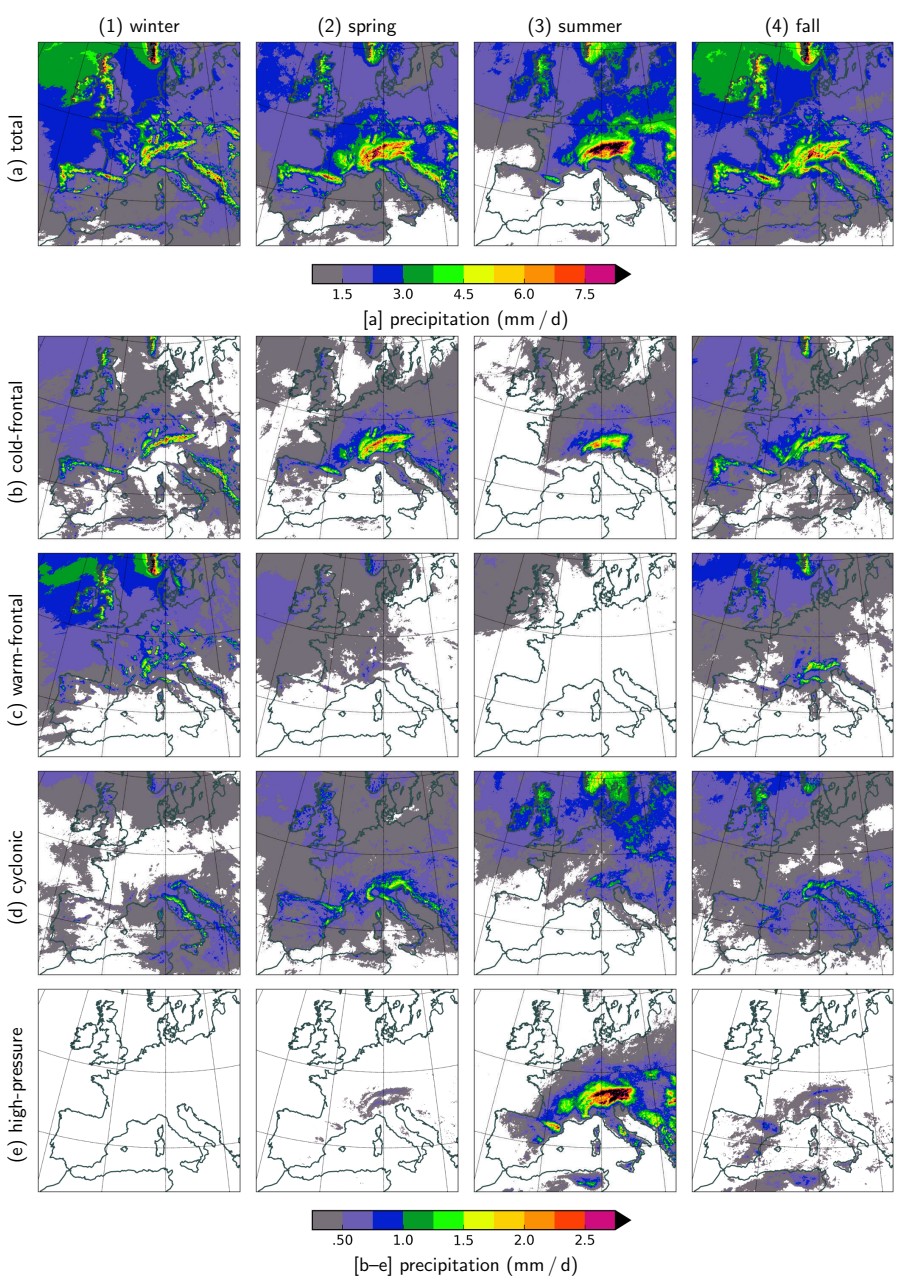

**Figure 8.** Mean daily precipitation during (1–4) each season of the nine-year period 2000–2008 (a) overall and (b–e) of select front-cyclone-relative contributions.

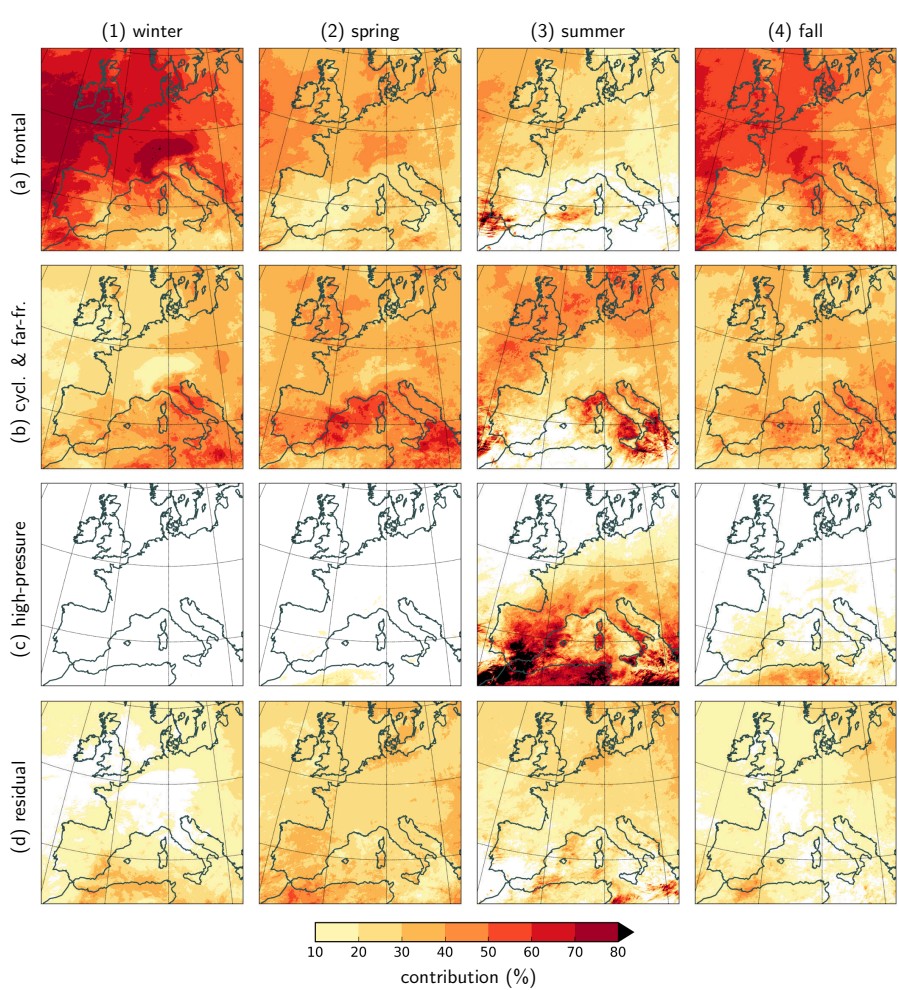

**Figure 9.** Relative precipitation contributions during (1–4) each season of the nine-year period 2000–2008 of front-cyclone-relative components: (a) sum of cold-frontal, warm-frontal, and collocated; (b) sum of cyclonic and far-frontal; (c) high-pressure; and (d) residual.

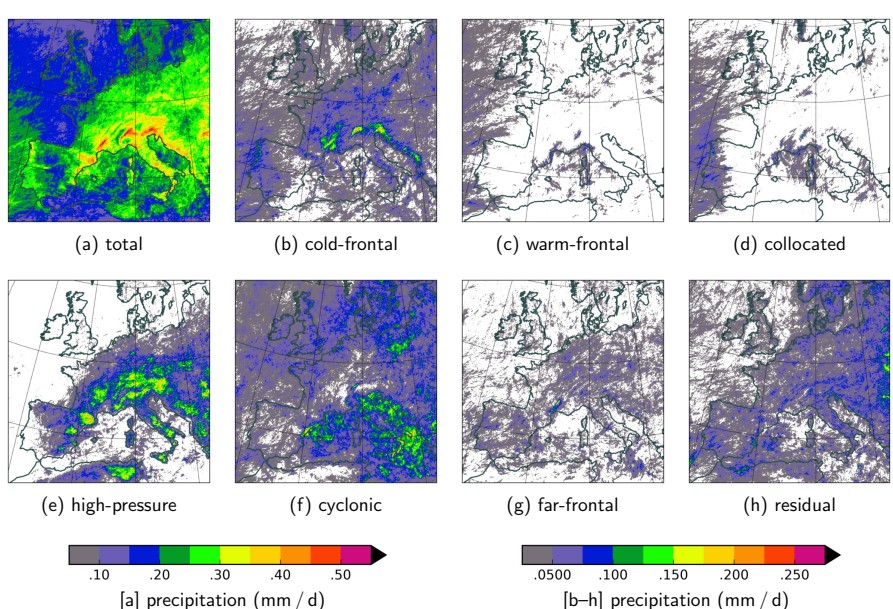

**Figure 10.** Like Fig. 7, but for annual heavy precipitation defined as the amount exceeding the local $99.9^{\text{th}}$ all-hour percentile of hourly precipitation intensity over the whole year.

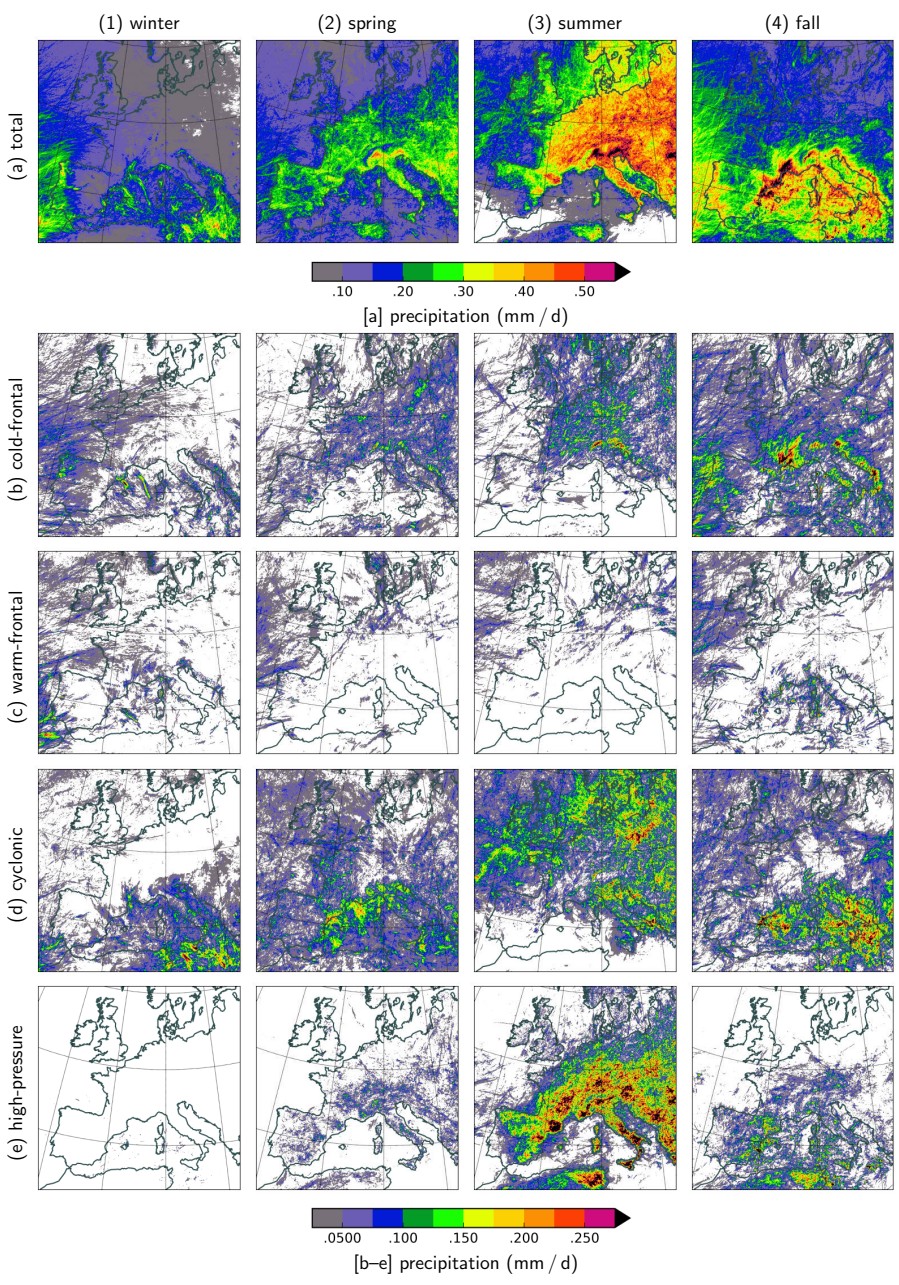

**Figure 11.** Like Fig. 8, but for seasonal heavy precipitation defined as the amount exceeding the local $99.9^{th}$ all-hour percentile of hourly precipitation intensity in a given season.

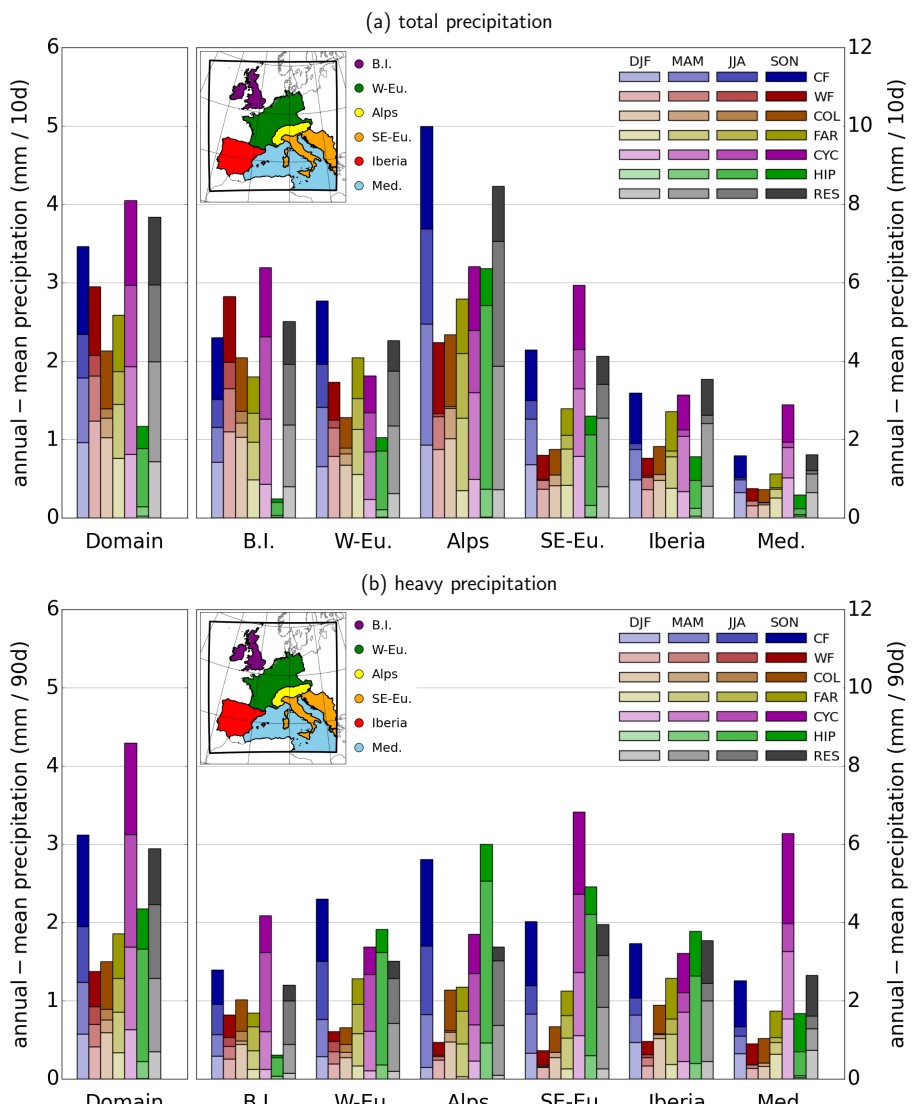

**Figure 12.** Mean (a) total and (b) heavy precipitation over the analysis domain and six select regions, as indicated in the map: British Isles, Western Europe, Alps, Southeastern Europe, Iberia, and the Mediterranean Sea. Heavy precipitation is defined as the amount of hourly precipitation above the local (grid-point specific) seasonal $99.9^{th}$ all-hour percentile. Each bar shows the annual-mean precipitation contribution of one front-cyclone-relative component (CF: cold-frontal; WF: warm-frontal; COL: collocated; FAR: far-frontal; CYC: cyclonic; HIP: high-pressure; RES: residual), with the four segments indicating the relative contribution of each season (DJF: winter; MAM: spring; JJA: summer; SON: fall). To obtain approximate absolute seasonal-mean amounts, multiply the height of a bar segment by four. Note that there is no relation between the colors of the bars and those of the regions on the map.





**Table 1.** Mid-monthly $|\nabla\theta_e|$ threshold values in K $(100\,\text{km})^{-1}$ to compute the thermal component of frontal areas, as described in Sec. 2.3.

| Jan | Feb | Mar | Apr | May | Jun | Jul | Aug | Sep | Oct | Nov | Dec |
|-----|-----|-----|-----|-----|-----|-----|-----|-----|-----|-----|-----|
| 4.0 | 4.0 | 5.0 | 6.0 | 7.0 | 8.0 | 8.0 | 8.0 | 7.0 | 6.0 | 5.0 | 4.0 |

**Table 2.** Mid-monthly $\Phi$ threshold values in $\text{m}^2\text{s}^{-2}$ to compute the $\Phi$-component of high-pressure areas at $850\,\text{hPa}$, as described in Sec. 2.4.

| Jan | Feb | Mar | Apr | May | Jun | Jul | Aug | Sep | Oct | Nov | Dec |
|------|------|------|------|------|------|------|------|------|------|------|------|
| 1550 | 1550 | 1525 | 1500 | 1475 | 1450 | 1450 | 1450 | 1475 | 1500 | 1525 | 1550 |