# Peer review of "Attribution of precipitation to cyclones and fronts over Europe in a kilometer-scale regional climate simulation"

_Weather and Climate Dynamics, 2020_

## Referee Comment (RC1) · Anonymous Referee #1 · 11 Jun 2020

This is a very timely paper on attribution of precipitation to main rain-bearing systems. It is not the first attempt to associated precipitation to various synoptic features, but this time it is more detailed and done using outputs of a convection-resolving model. I also like that results show annual and seasonal data for all 4 seasons, as there are important seasonal differences. The manuscript features excellent literature review and is well written.

I have relatively minor comments listed below. I am most concerned about attributing precipitation to high pressure systems. As the authors say, it is most likely associated with convection, so I made a few suggestions on that in the comments. Another suggestion is to add a threshold on the size of frontal areas, as there are many very small frontal features in the examples. Finally, I am interested if similar approaches are applied to ERA5(or other reanalyses), how the results will be different. The latter might be outside the scope of this paper, so I wish to see such comparison some time in the future.

Commnets:

158, 185: The 12 km domain is not significantly larger than 2.2km domain. Did you consider merging with ERA-interim? It might be particularly good for getting cyclones and high pressure systems right.

193-194: this not clear. Please explain better what you mean by allowing 20% of contours to cross the boundary before 'halting further feature growth'.

l.200: in the abstract it is said that local thermal fronts are removed, here you say that fronts are categorised at synoptic and local. Are local fronts removed then?

215: what is the threshold value on theta e gradient based on and why all values in Table 1 are whole numbers?

l.252: it is not clear to me how high-pressure systems are defined. One may think that you mean anticyclones (i.e. an area similar to cyclones with high pressure in the middle circled by a closed contour), but 'high pressure' systems in fig. 3 look confusing. In fig 3 (summer) the green area looks like the subtropical ridge (there are big and small white areas within green stippling - what do they represent?), in fig 5 (winter) I would suspect an anticyclone defined using the MSLP field. These systems need to be better described, both their identification procedure and physical meaning. There is a recent paper by Poujol et al. (2020) on a separation between convective and stratiform precipitation. It might be interesting to check if the precipitation within high pressure systems can be classified as convective using their approach. Discussion around lines 400 and 463 may benefit if you mention possible convective nature of

high pressure precipitation, that is prevalent in summer. Given the frequency of high-pressure 'components' (fig. S4), which cover 50% of your domain 50% of time in summer, these systems need to be explored in more detail.

fig.3: The example is very good, but I have numerous suggestions on plotting: The red outline stands for local fronts, while red filling (in slightly different shade) - for warm fronts. It would be good to use different colours. A bold black contour also circles the cyclone area, is that right? I think it is not mentioned int he caption. Blue filling of cold fronts is very similar to precipitation 0.2-1mm/h, please use different colours. I am not sure I can see red filling well for the warm front (it works better in fig. 5). Is warm front in fig. 3 a 'local' front, not synoptic? If this is the case then the separation between local and synoptic fronts is probably not working very well. It would be good to remind the reader that frontal systems within the high pressure system do not count as rain-bearing (i.e. this precipitation is attributed to the high pressure system only).

Fig. 3 makes me think that it would be good to have a threshold on the size of the frontal area to remove very small features.

l.412, Fig8 vs fig 9, high pressure precipitation: In figure 8 high-pressure precipitation is over the land only (with an exception for the Bay of Biscay), but for relative precipitation there is a large proportion of convective precipitation over the Mediterranean sea. Can you explain this?

Fig.9: I find it odd that cyclone and far-frontal precipitation are combined in this plot. I am not sure if this information is valuable. Is it possible to separate them?

l.438: Are you able to explain high amount of residual heavy precipitation comparable to, e.g., cold-frontal heavy rainfall?

Minor comments:

fig. 1 and possibly other plots: Please add lon/lat values.

p.2,l. 38: ", high-pressure systems, extratropical cyclones, fronts, orography ... contribute to precipitation" - I'd avoid starting with high pressure systems as they not the main rain-bearing systems

p.2, l45; p3.l.78, 80: I think it should read "such resolution", "such attribution"

p.2, l47: I'd rather say "interplay between fronts and steep orography in producing precipitation"

p.2, l53-55: I doubt this sentence is needed

l.98: re-phrase 'on a continental-scale domain'; perhaps, 'for a continental-size domain' or "on a scale of a continent'

l.114: ' domain covering most of Europe' - I disagree, though it is hard to get the area by eye. Given the size of Eastern Europe (former USSR seems to be excluded from analysis) and Scandinavian counties, my feeling is that the domain covers roughly half of Europe.

l.124:"this attribution" replace with "their contribution"

133: 'can be found' instead of 'is found'

156: replace interpolate with extrapolate

165: "the features are interpolated back onto the original 2.2 km grid". I do not think this is the right way of describing it. My understanding is that you first create a mask based on 12 km field and then use it on 2.2 km scale.

392: change to 'selected'

l.558 and in thought the manuscript: I would avoid saying that summer precipitation is 'associated' with high-pressure systems, though technically this is what the paper shows. As you say, it is most likely associated with convection. I'd rather say that summer precipitation is often detected within high pressure systems.

fig.S2: Why do you need 'track frequencies', would simply 'frequencies' not be enough?

fig.S4: components of what?

 References:

Poujol, B, Sobolowski, S, Mooney, P, Berthou, S. A physically based precipitation sep- aration algorithm for convection‐permitting models over complex topography. Q J R Meteorol Soc. 2020; 146: 748– 761. https://doi.org/10.1002/qj.3706

---

## Referee Comment (RC2) · Anonymous Referee #2 · 16 Jun 2020

The comment was uploaded in the form of a supplement:
https://wcd.copernicus.org/preprints/wcd-2020-18/wcd-2020-18-RC2-supplement.pdf

———————————————

---

## Author Response (AR1)

Paper wcd-2020-18

**Attribution of precipitation to cyclones and fronts over Europe in a kilometer-scale regional climate simulation**

Stefan Rüdisühli, Michael Sprenger, David Leutwyler, Christoph Schär, and Heini Wernli

**Response to the Reviewers' comments:**

We thank both reviewers for their constructive and helpful comments that helped to further improve the presentation of our results.

Following are detailed replies to the individual comments.

**Reviewer #1**

**This is a very timely paper on attribution of precipitation to main rain-bearing systems. It is not the first attempt to associated precipitation to various synoptic features, but this time it is more detailed and done using outputs of a convection-resolving model. I also like that results show annual and seasonal data for all 4 seasons, as there are important seasonal differences. The manuscript features excellent literature review and is well written.**

Many thanks for these positive statements!

**I have relatively minor comments listed below. I am most concerned about attributing precipitation to high pressure systems. As the authors say, it is most likely associated with convection, so I made a few suggestions on that in the comments. Another suggestion is to add a threshold on the size of frontal areas, as there are many very small frontal features in the examples. Finally, I am interested if similar approaches are applied to ERA5 (or other reanalyses), how the results will be different. The latter might be outside the scope of this paper, so I wish to see such comparison sometime in the future.**

We address all these points below.

**Comments:**

**1. l.158, 185: The 12 km domain is not significantly larger than 2.2 km domain. Did you consider merging with ERA-interim? It might be particularly good for getting cyclones and high-pressure systems right.**

While indeed the difference in domain size between the 2.2 km and the 12 km simulation is not huge, it still makes a large difference for the cyclone identification because the influence of the domain boundary on cyclone feature growth is largest in the direct vicinity of the boundary.

In fact, we have previously investigated the influence of the domain boundary on the cyclone identification (see Rüdisühli (2018), https://doi.org/10.3929/ethz-b-000351234), comparing cyclones identified in ERA-Interim (on a global grid), the 12 km simulation, and the 12 km simulation with a reduced domain size (minus 40 grid points in all directions, corresponding to the domain of the 2.2 km simulation, but without including any 2.2 km data). The absolute cyclone frequencies are shown in the following figure:

[Figure]

Note that the absolute frequencies between ERA-Interim and the 12 km simulation are not directly comparable, as we did not account for differences in grid resolution etc. Also note that differences in feature frequency are largely caused by differences in the mean feature size rather than differences in the occurrence frequency of the features. In other words, the 12 km simulation has a similar number of cyclones as ERA-Interim, e.g., near Scotland, however they are smaller (because of the limited domain) and therefore the frequencies are lower.

But most relevant here is the comparison of the two 12 km composites. It is obvious that the reduced domain size has a large impact on the cyclones over the North Atlantic, whereas the frequencies are very similar over the southern half of the domain. The influence of the boundary becomes even more obvious if we look at the ratio of the above cyclone frequency fields from the 12 km simulation:

[Figure]

The figure clearly reveals the strong boundary influence, as the cyclones very rapidly decrease in size (and thus composite frequency) very close to the boundaries.

These comparisons give an idea of the benefit of increasing the domain size, in particular for the northern part of the model domain where cyclones typically propagate rapidly across the domain. Properly tuned, increasing the domain size of the 12 km domain using ERA-Interim – as suggested by the reviewer – may have a similar effect as increasing the 12 km domain from reduced to full size, with a large effect in a narrow boundary zone and a smaller effect in the interior of the domain.

In conclusion: Yes, there would likely be a benefit on the identified cyclones by increasing the domain further with ERA-Interim, but already the relatively small increase from the 2.2 km to the 12 km domain has a substantial positive effect on the cyclones in the analysis domain, removing the worst of the boundary effect (as shown above). This is why we decided that the additional effort of incorporating ERA-Interim data was not worth it; while certainly not perfect, for our purposes (of distinguishing the zone close to the cyclone center from the rest of a cyclonic system) the identified cyclones are good enough.

For our high-pressure areas, on the other hand, extending the domain further would not make any difference, as they are based on the local values of the geopotential field and its gradient (see also our answer to comment 5).

**2. l.193-194: This not clear. Please explain better what you mean by allowing 20% of contours to cross the boundary before 'halting further feature growth'.**

We agree and have changed the text to express this more clearly.

*Old:*
*[...] We opt for a compromise by allowing one in five contours of a feature (20 %) to cross the boundary before halting further feature growth.*

*New:*
*[...] We opt for a compromise by allowing up to 20 % of the contours of a feature to be boundary crossing. For example, if 16 closed contours are identified around a pressure minimum before the boundary is reached, then at most four additional boundary-crossing contours can be added before the 20 % threshold is reached at four out of twenty contours.*

**3. l.200: In the abstract it is said that local thermal fronts are removed, here you say that fronts are categorised at synoptic and local. Are local fronts removed then?**

Yes, the local fronts are removed for this analysis. We have added a sentence to emphasize this.

*New:*
*The local fronts are then removed and only the synoptic fronts are used in this study.*

**4. l.215: What is the threshold value on theta-e gradient based on and why all values in Table 1 are whole numbers?**

This is a good question, pointing to the challenge of reasonably choosing the theta-e gradient thresholds. To the best of our knowledge, there is no fully objective procedure to determine these thresholds. We found the monthly threshold values subjectively by examining multiple years of data. Specifically, we have evaluated the fronts based on a range of possible thresholds, deduced monthly "best estimates" based on how well the front features matched the meteorological fields, and based on these determined the thresholds listed in Table 2. Given their subjective and approximate nature, and their magnitude, there was no reason not to settle on whole numbers. We mention this challenge of choosing appropriate thresholds also in the third bullet point of our conclusions.

**5. l.252: It is not clear to me how high-pressure systems are defined. One may think that you mean anticyclones (i.e. an area similar to cyclones with high pressure in the middle circled by a closed contour), but 'high pressure' systems in fig. 3 look confusing. In fig 3 (summer) the green area looks like the subtropical ridge (there are big and small white areas within green stippling - what do they represent?), in fig. 5 (winter) I would suspect an anticyclone defined using the MSLP field. These systems need to be better described, both their identification procedure and physical meaning. There is a recent paper by Poujol et al. (2020) on a separation between convective and stratiform precipitation. It might be interesting to check if the precipitation within high pressure systems can be classified as convective using their approach. Discussion around lines 400 and 463 may benefit if you mention possible convective nature of high-pressure precipitation, that is prevalent in summer. Given the frequency of high-pressure 'components' (fig. S4), which cover 50% of your domain 50% of time in summer, these systems need to be explored in more detail.**

The identification procedure of the high-pressure areas is described in Sec. 2.4: They are areas with high pressure ($\Phi$ at 850 hPa above threshold derived from monthly values in Table 2, which have been found by a similar subjective evaluation as the frontal gradient thresholds in Table 1) and a flat pressure distribution ($\nabla\Phi < 0.02$ m s$^{-2}$). As opposed to fronts and cyclones, the high-pressure areas are simple masks, without any sophisticated feature identification or tracking. The white areas within the green stippling in Fig. 3 are therefore regions where either the geopotential is locally too low and/or its gradient is locally too strong.

We agree that the motivation, physical meaning, and name of the high-pressure areas are not explained in sufficient detail. We have revised and extended Sec. 2.4 to more clearly convey these points.

*Old:*
*Precipitation not only occurs near cyclones and fronts, but also in areas of weak synoptic forcing typically characterized by relatively high pressure or by a flat pressure distribution, for example with diurnal summer convection over the continent. We explicitly identify such high-pressure areas based on geopotential $\Phi$ and its gradient $\nabla\Phi$ at 850 hPa. Seasonal feature frequency composites are provided in the supplementary material (Fig. S1).*
*The $\Phi$ field is first smoothed with a Gaussian filter. A mask is derived by applying a minimum threshold that varies over the year to account for the seasonal cycle in $\Phi$. Analogous to the seasonally varying frontal threshold, the $\Phi$ threshold values are defined in the middle of each month (Table 2) and linearly interpolated to each hour in-between. Then, $\nabla\Phi$ is computed, and the resulting field is smoothed again. A second mask is derived by applying a constant maximum threshold of 0.02ms−2 to $\nabla\Phi$. The high-pressure area corresponds to the overlap area of the $\Phi$ and $\nabla\Phi$ masks. All threshold values have been determined subjectively based on thorough manual testing.*

*New:*
*Precipitation not only occurs near cyclones and fronts, but also in areas of weak synoptic forcing typically characterized by relatively high pressure and a flat pressure distribution, for example with diurnal summer convection over the continent. When attributing precipitation only to cyclones and fronts, such precipitation would not be captured and become part of the residual. Our original method without high-pressure areas, however, often misclassified diurnal summer convection as front-related (specifically far-frontal, as defined in Sec. 2.5). To prevent this, we explicitly identify such areas characterized by high pressure and a flat pressure distribution – henceforth simply called high-pressure areas – based on the geopotential $\Phi$ and its gradient $\nabla\Phi$ at 850 hPa. Seasonal frequency fields of the identified high-pressure areas are provided in the supplementary material (Fig. S1).*
*Computing the high-pressure areas at 850 hPa involves the following steps:*
1. *Smooth the $\Phi$ field using a Gaussian filter with a standard deviation $\sigma=3$. Then compute a $\Phi$ mask covering areas with high pressure, based on a minimum threshold, which varies over the year to account for the seasonal cycle in $\Phi$. The threshold at a given time step is derived by linear interpolation from the mid-monthly threshold values listed in Table 2.*
2. *Smooth the $\Phi$ field again using a Gaussian filter with a standard deviation $\sigma=20$, then compute $\nabla\Phi$, whereby the gradient at each grid point is computed across multiple unit grid distances using offsets of ($i\pm10$, $j\pm10$), corresponding to $\pm120$ km in our hybrid 12 km fields. Then compute a $\nabla\Phi$ mask covering areas with a weak pressure gradient, based on a constant maximum threshold of 0.02 m s$^{-2}$.*
3. *The high-pressure area corresponds to the overlap area of the $\Phi$ and $\nabla\Phi$ masks.*
*All threshold values have been determined subjectively based on extensive manual evaluation of multiple years of data.*

We thank the reviewer for pointing us to the paper by Poujol et al. (2020), as their classification approach looks very promising. However, we do not think that it would add much to the characterization of the high-pressure area precipitation, because it is fairly obvious to us (from extensive visual analysis of precipitation fields during method development) that most of this

precipitation is convective. But their separation of precipitation types could be a great extension of our attribution method, which could be addressed future studies. We have added such a remark to the end of the "Conclusions".

*New:*
*Finally, methods that separate precipitation types like convective and stratiform (e.g., Poujol et al., 2020) could be combined with our feature-based attribution, which would enable a more in-depth characterization of the different front-cyclone-relative precipitation components*

*References:*

*Poujol, B., Sobolowski, S., Mooney, P., and Berthou, S.: A physically based precipitation separation algorithm for convection-permitting models over complex topography, Q.J.R. Meteorol. Soc., 146, 748–761, https://doi.org/10.1002/qj.3706, 2020.*

**6. Fig.3: The example is very good, but I have numerous suggestions on plotting:**
- **The red outline stands for local fronts, while red filling (in slightly different shade) - for warm fronts. It would be good to use different colours.**
- **A bold black contour also circles the cyclone area, is that right? I think it is not mentioned in the caption.**
- **Blue filling of cold fronts is very similar to precipitation 0.2-1 mm/h, please use different colours.**

We agree that the plotting is not optimal, especially with respect to the color clashes. In hindsight, we tried to achieve too much at the same time: intuitive colors for cold and warm fronts (blue for cold, red for warm) and for synoptic and local fronts (black for "good", red for "bad"), while still sticking with the same precipitation color map as in all other figures – which, unfortunately, also includes red and blue, as the reviewer has pointed out.

We have therefore adapted the figures as follows:
- Precipitation is shown in shades of gray, which prevents color clashes.
- Cold and warm fronts are now distinguished by the color of their outlines (blue and red), while the filling has been removed because it clashed with the precipitation field, one necessarily obscuring the other.
- Local fronts are now all highlighted by the same outline color (orange), because the type of the local fronts is of secondary importance at best.
- (For Uriah, show the time steps in the figure that are actually described in the text.)

Regarding the caption, cyclones were actually mentioned (*"[...] the unfilled bold contours the outlines of cyclone features; [...]"*), but we concede that the sentence was not easy to read. When rewriting the caption, we have tried to make it more easily understandable.

[Figure]

*Old:*
*Figure 3. Development of cyclone Uriah in June 2007. The thin black contours indicate geopotential at 850 hPa; the colored shading the surface precipitation; the filled bold contours the outlines of front features, with black/red outlines for synoptic/local fronts, and blue/red filling for cold/warm fronts; the unfilled bold contours the outlines of cyclone features; and the green stippling high-pressure features.*

*New:*
*Figure 3. Development of cyclone Lancelot in January 2007. Thin black contours indicate the geopotential at 850 hPa, gray shading the surface precipitation, and green stippling the high-pressure areas. Bold contours represent the outlines of tracked features: synoptic cold and warm fronts (blue and red), local fronts of either type (orange), and cyclones (black).*

- **I am not sure I can see red filling well for the warm front (it works better in fig. 5). Is warm front in fig. 3 a 'local' front, not synoptic? If this is the case then the separation between local and synoptic fronts is probably not working very well.**

Indeed, the warm front in this case is too weak to be robustly classified as a synoptic front. For this particular case, one might thus argue that the classification is not working very well. However, one must keep in mind that the algorithm has not been tuned for this specific case, but such that it does a reasonable job in the majority of cases. We have chosen this June case because it meteorologically contrasts the January case in many respects (slow instead of fast, dominant cold front instead of pronounced warm front, "Norwegian" instead of "Shapiro-Keyser", summer instead of winter), without too much consideration of the performance of the identification algorithm. Our goal is to illustrate the performance of the algorithm in representative – albeit meteorologically attractive – scenarios, rather than "cherry-pick" cases where the algorithm does an especially good job.

That being said, it is indeed not optimal that there is no synoptic warm front in the Figure of the summer case, given they should still be mentioned in the caption. We have therefore decided to switch the two case studies, such that this type of figure can be introduced for the winter case (in which all elements are present) and then referred to by the respective summer case figure.

We have also adapted the text to stress that the warm front is not detected as a synoptic feature by the algorithm and thus not used for the precipitation attribution.

- **It would be good to remind the reader that frontal systems within the high-pressure system do not count as rain-bearing (i.e. this precipitation is attributed to the high-pressure system only).**

We agree and have added a short remark to the caption of Fig. 5: *"Note that in (a), the precipitation along the cold front over northwestern Spain will be attributed to the high-pressure area instead, which takes precedence over fronts (see Sec. 2.5)."*

**7. Fig. 3 Makes me think that it would be good to have a threshold on the size of the frontal area to remove very small features.**

We definitely tried that. The problem with an explicit feature size threshold is that it harms as much as it helps. While it would surely remove some spurious features that we'd prefer to get rid of, it would also remove many that we do want to retain, for instance fronts associated with small cyclones over the Mediterranean, or fragments of large fronts that are not connected to the main feature. (Fragmentation is fairly common for all but the largest fronts, given most of our domain is over land.)

We tried many different approaches and combinations of criteria, and finally settled on the two criteria described in Sec. 2.3: "typical feature size" and "stationarity". The former does indeed consider feature size, but for whole tracks rather than individual time steps, which makes it more robust in case of fragmentation. The stationarity criterion allows for, e.g., the mobile fronts associated with small Mediterranean cyclones to be classified as non-local and thus make it into the analysis. Fig. 3 actually illustrates that these criteria work fairly well, as most small-scale features are classified as local (red outlines) while the larger, precipitating fronts are classified as synoptic.

**8. l.412, Fig 8 vs Fig 9, high pressure precipitation: In figure 8 high-pressure precipitation is over the land only (with an exception for the Bay of Biscay), but for relative precipitation there is a large proportion of convective precipitation over the Mediterranean Sea. Can you explain this?**

Yes, we can. Fig. 8 shows absolute precipitation amounts of the components, starting at 0.25 mm/d, while Fig. 9 shows the relative contributions of the components to the absolute precipitation amount. In summer, there is hardly any precipitation at all over the Mediterranean Sea: less than 0.75 mm/d overall, and none of the four shown components exceeds 0.25 mm/d. However, as Fig. 9 shows, there is some precipitation, and a substantial fraction of it is associated with high-pressure areas (i.e., presumably convective). In fall, on the other hand, there is substantially more precipitation over the Mediterranean Sea than in summer with about 1.5 mm/d on average, and the high-pressure contributions locally exceed 0.25 mm/d in several places. The modest about 10% high-pressure contributions in Fig. 9 are thus consistent.

**9. Fig. 9: I find it odd that cyclone and far-frontal precipitation are combined in this plot. I am not sure if this information is valuable. Is it possible to separate them?**

Yes, it is possible to separate them. We opted to combine them to reduce the number of plots; the focus of the figure is mainly on the other contributions (frontal, high-pressure, residual), so we combined cyclonic and far-frontal into "other front/cyclone-related".

Upon reflection, we do agree that this is probably not the most meaningful way to combine these groups. We have therefore separated the far-frontal and cyclonic contributions, now showing them separately.

**10. l.438: Are you able to explain high amount of residual heavy precipitation comparable to, e.g., cold-frontal heavy rainfall?**

The residual is especially large in spring, which is when fronts (especially warm fronts) already occur less frequently than during their peak in winter, but high-pressure areas have not yet reached their peak frequency in summer. Given the larger residual heavy precipitation in spring compared with the other seasons, especially over land, some of this may be due to early convective precipitation events, which are not captured by our high-pressure areas to the degree they are in summer.

Similarly, over Sweden, the large amounts of heavy residual precipitation in spring coincide with a lower cyclone frequency compared with summer. Possibly, convective precipitation events in spring are triggered by other processes than cyclones, while in summer, many are associated with cyclones when those occur with high frequency.

In summer, the residual is distributed fairly evenly across the domain and roughly comparable to the total frontal contributions, while the cyclonic and high-pressure contributions are substantially larger. This is not surprising given the frequency minimum in both cold and warm fronts in summer.

In fall, residual contributions are relatively large over the Baltic states, where the front and cyclone frequencies are much lower than further west. In addition, this is close to the upper-left corner of the domain, so in addition to natural decay of these systems, boundary effects on the feature identification may also play some role.

Finally, from fall through spring residual precipitation is relatively frequent and heavy along the North African coast. Only few cyclones and fronts occur in this region, and the high-pressure area frequency is also much lower than in summer, so most precipitation is classified as residual. However, since this region is drier than most areas further north, large relative residual contributions still translate to relatively little residual precipitation in absolute terms.

**Minor comments:**

**11. Fig. 1 and possibly other plots: Please add lon/lat values.**

We agree that the grid lines should be labeled in Fig. 1. We have redone this figure with grid line labels, and in the process also added the inner boundary of the blending zone.

As for the other plots, we are of the opinion that grid line labels do not offer much benefit (the location of the domain is obvious given the European coastlines, the shown grid lines are only major and thus easily deduced from the coastlines, and the domain is the same in all plots), but major downsides (if placed in the plots, it would fill them up even more and make it even harder to deduce details as it already is given their small size, while if placed outside the plots, it would increase the size of the multi-panel figures, potentially necessitating even smaller maps).

[Figure]

*Old:*
*Figure 1. COSMO simulation domains and model topography. The outermost black box denotes the domain of the convection-parameterizing simulation with a grid spacing of 12 km and the bold box the domain of the convection-resolving simulation with 2.2 km grid spacing. The innermost thin box indicates the subdomain used in the analysis. (Figure and caption from Leutwyler et al., 2017)*

*New:*
*Figure 1. Domain boundaries and model topography of the two COSMO simulations. The four black boxes show, from large to small: (bold) the model domain of the driving simulation with a horizontal grid spacing of 12 km; (semi-bold) the model domain of the nested simulation with a horizontal grid spacing of 2.2 km; (thin) the subdomain of the 2.2 km domain on which the precipitation attribution analysis is performed; and (dashed) the inner boundary of the blending zone that is used during the computation of the hybrid fields on which the feature identification is based (see Sec. 2.1). The model topography inside (outside) the 2.2 km domain boundary is that of the nested 2.2 km(driving 12 km) simulation*

**12. l38: ", high-pressure systems, extratropical cyclones, fronts, orography ... contribute to precipitation" - I'd avoid starting with high pressure systems as they are not the main rain-bearing systems**

We agree and have changed it to "[...] extratropical cyclones, fronts, orography, high-pressure systems, and their interactions [...]".

**13. l45, l.78, 80: I think it should read "such resolution", "such attribution"**

We agree and have changed it as proposed.

**14. l47: I'd rather say "interplay between fronts and steep orography in producing precipitation"**

We agree and have changed it as proposed.

**15. l53-55: I doubt this sentence is needed**

We agree and have removed the sentence.

**16. l.98: Re-phrase 'on a continental-scale domain'; perhaps, 'for a continental-size domain' or "on a scale of a continent'**

We agree this could be phrased better and have changed it to "at a continental scale".

**17. l.114: 'domain covering most of Europe' - I disagree, though it is hard to get the area by eye. Given the size of Eastern Europe (former USSR seems to be excluded from analysis) and Scandinavian counties, my feeling is that the domain covers roughly half of Europe.**

We agree that the domain does not cover Eastern Europe, but Western Europe and a large fraction of the Mediterranean. We have therefore changed "the comparatively large domain covering most of Europe" to "the decade-long simulation on a computational domain capable of representing the evolution of these systems over Western Europe, the eastern North Atlantic, and the Mediterranean".

We note that the computational domain covers an area of about 11,000,000 km$^2$, which is a bit larger than the European land area.

**18. l.124: "this attribution" replace with "their contribution"**

We partially agree and have changed it to "these contributions".

**19. l.133: 'can be found' instead of 'is found'**

We agree and have changed it as proposed.

**20. l.156: Replace interpolate with extrapolate**

Given this transformation step is only performed over the part of the 12 km grid covered by the 2.2 km grid, where we have data, we think that "interpolate" is indeed the right word. However, we concede that the whole explanation of the procedure could be clearer (see also next comment) and have thus rephrased it.

**21. l.165: "the features are interpolated back onto the original 2.2 km grid". I do not think this is the right way of describing it. My understanding is that you first create a mask based on 12 km field and then use it on 2.2 km scale.**

Indeed, the feature masks are first created on the 12 km grid based on the "hybrid fields" and then used at 2.2 km scale to attribute the precipitation fields from the 2.2 km simulation to the features. Technically, this involves interpolating the feature masks from the 12 km to the 2.2 km grid (where the data in the interior of the domain has originally come from, thus the "back"). We have rephrased the explanation of the whole procedure (see also previous comment) to make it clearer and more precise.

*Old:*
*[...] In order to exploit the advantages of both simulations, the 2.2 km and 12 km data are merged in the following three-step procedure:*
   1. *Interpolate the 2.2 km fields onto the 12 km grid. This retains the exact position and extent of the cyclones and fronts in the 2.2 km simulation while increasing the signal-to-noise ratio to the level of the 12 km simulation.*
   2. *Paste these into the 12 km fields to obtain hybrids comprised of 2.2 km simulation data in the center and 12 km simulation data beyond the boundaries of the inner nest.*
   3. *Introduce a blending zone along the boundaries in the inner domain with a smooth transition from the 2.2 km data to the 12 km data. It extends 50 coarse grid points (~60 km) into the inner domain and is based on the logistic function $1/(1 + exp−k×x)$ with $k= 0.8$.*
*This retains the exact position and extent of the cyclones and fronts in the 2.2 km simulation while increasing the signal-to-noise ratio to the level of the 12 km simulation. The resulting hybrid fields reside on the grid of the 12 km simulation and thus benefit from its large domain and relatively low noise level, while being meteorologically consistent with the 2.2 km simulation within the analysis domain in the inner nest. We use them to identify cyclones (Sec. 2.2) and fronts (Sec. 2.3). Before conducting the precipitation attribution analysis (Sec. 2.5), however, the features are interpolated back onto the original 2.2 km grid.*

*New:*
*[...] In order to exploit the advantages of both simulations, the 2.2 km and 12 km data are merged in the following procedure:*
   1. *Interpolate the 2.2 km fields to the part of the 12 km grid covered by the domain of the 2.2 km simulation.*
   2. *In the interior of the domain at a distance of at least 50 coarse grid points (~600 km) from the boundary of the 2.2 km domain, use these fields from the 2.2 km simulation.*
   3. *Outside the 2.2 km domain, use the fields from the 12 km simulation.*
   4. *In-between, blend the fields with $f= 0.1/(1 + exp (−0.8 × (10x − 5)))$, where $x$ increases linearly from 0.0 at the inner boundary of the blending zone to 1.0 at the outer boundary and f increases logistically in the same range, corresponding to the fraction of 12 km data*
*The resulting hybrid fields possess the bigger domain and lower noise level of the 12 km simulation, which allows for a more robust feature identification over the analysis domain, especially close to the boundaries such as over the North Atlantic. At the same time, the hybrid fields are meteorologically consistent with the 2.2 km simulation.*

*We use the hybrid fields on the 12 km grid to identify cyclones (Sec. 2.2), fronts (Sec. 2.3), and high-pressure areas (Sec. 2.4), and then use the resulting feature masks at 2.2 km for the precipitation attribution analysis (Sec. 2.5).*

**22. l.392: Change to 'selected'**

We agree and have changed it as proposed.

**23. l.558 and throughout the manuscript: I would avoid saying that summer precipitation is 'associated' with high-pressure systems, though technically this is what the paper shows. As you say, it is most likely associated with convection. I'd rather say that summer precipitation is often detected within high pressure systems.**

We agree and have adapted this sentence accordingly.

*Old:*
*It is interesting that this approach confirms the strongly opposing character of winter and summer precipitation, the former being very strongly associated with cyclones and fronts, and the latter predominantly with high-pressure systems.*

*New:*
*It is interesting that this approach confirms the strongly opposing character of winter and summer precipitation, the former being very strongly associated with cyclones and fronts, the latter predominantly detected within high-pressure systems.*

**24. Fig. S2: Why do you need 'track frequencies', would simply 'frequencies' not be enough?**

No, "frequencies" would be ambiguous. As explained in the respective caption, the "track frequencies" are computed by compositing the track masks (which comprise all grid points that have encountered at least one feature belonging to the track at least once), as opposed to the complementary "feature frequencies" (e.g., Fig. S1), for which all individual feature masks are composited.

**25. Fig. S4: Components of what?**

"Front-cyclone-relative" components, as in Fig. S3. The domain is separated at each time step into seven masks corresponding to these components (before these masks are applied to the precipitation field). Figs. S3 and S4 show frequency composites of these masks. We have adapted the Figure captions to express this more clarity.

*Old:*
*Figure S3. Frequencies of front-cyclone-relative components during (0) the whole year, (1) winter (DJF), (2) spring (MAM), (3) summer (JJA), and (4) fall (SON) 2000–2008. Shown are the (a) cold-frontal, (b) warm-frontal, (c) collocated, and (d) far-frontal components.*
*Figure S4. Like Fig. S3, but showing the frequencies of the (e) cyclonic, (f) high-pressure, and (g) residual components.*

*New:*
*Figure S3. Frequencies of front-cyclone-relative component masks during (0) the whole year, (1) winter (DJF), (2) spring (MAM), (3) summer (JJA), and (4) fall (SON) 2000–2008. The masks are obtained at each time step by*

*separating the domain into seven components as described in Sec. 2.5. Shown are the (a) cold-frontal, (b) warm-frontal, (c) collocated, and (d) far-frontal components.*

*Figure S4. Frequencies of front-cyclone-relative component masks as in Fig. S3, but showing the (e) cyclonic, (f) high-pressure, and (g) residual components.*

The reason why the criteria focus on the local fronts is that the primary motivation to introduce this distinction in the first place was to remove the local fronts from the data set as they were so abundant, especially before we switched from using the 2.2 km data to the hybrid fields on the 12 km grid. However, we agree that it would indeed be more intuitive to focus the grouping on the synoptic rather than the local fronts, and have adapted the text accordingly. (We have also inverted the definition of *stationarity* such that it increases, rather than decreases, with higher values.)

*Old:*
*[...] Local fronts – largely produced by differential heating along topography and coasts – are generally smaller and more stationary than synoptic fronts. These properties can be expressed by a pair of criteria (on which we have settled after extensive manual testing):*

- *The* typical feature size *of a track is calculated by first combining, at each time step, the sizes of all features that belong to the track; and then calculating the median of these total sizes over all time steps. Front tracks are considered local if the* typical feature size *does not exceed 1000 km$^2$.*
- *The* stationarity *of a track is determined as its total footprint area (defined by all grid points that belong to the tracked front at any time) divided by the* typical feature size*. Front tracks are considered local if the* stationarity *does not exceed 6.0.*

*All tracks fulfilling one or both criteria are considered local fronts, and thus small and/or stationary. All remaining tracks are considered synoptic fronts, and thus both large and non-stationary.*

*New:*
*[...] Synoptic fronts are generally larger and more mobile (i.e., less stationary) than local fronts, which are largely produced by differential heating along topography and coasts. These properties can be expressed by a pair of criteria (on which we have settled after extensive manual testing):*

- *The* typical feature size *of a track is calculated by first combining, at each time step, the sizes of all features that belong to the track; and then calculating the median of these total sizes over all time steps. Front tracks are only considered synoptic if the typical feature size is at least 1000 km$^2$.*
- *The* stationarity *of a track is determined as the typical feature size divided by the total footprint area (defined by all grid points that belong to the tracked front at any time). Front tracks are only considered synoptic if the stationarity is below 0.167.*

*All tracks fulfilling both criteria are considered synoptic fronts, and thus both large and mobile. All remaining tracks are considered local fronts, and thus small and/or stationary. Only synoptic fronts are used for the precipitation attribution analysis, while local fronts are removed.*

**10. Line 267. For the far-frontal precipitation, do these features need to be also within a cyclone mask, or are both local and synoptic fronts included in this classification?**

Local fronts are not included in the analysis, only those fronts classified as synoptic. We have added a sentence stating this explicitly at the end of Sec. 2.3 (see answer to comment #9).

The front-cyclone-relative components are defined in the order listed in Sec. 2.5. The far-frontal component is defined second-to-last before only the residual, and thus does not include any grid points that have already been assigned to any other component, including the cyclonic.

(We're assuming that by "features", the reviewer is referring to "precipitation features", rather than front or cyclone features.)

**11. Line 273. During the subjective evaluation of the distance thresholds, was any seasonality identified? I.e. did similar thresholds capture the frontal precipitation in both winter and summer?**

We did not specifically evaluate the seasonality of the distance of the precipitation to the fronts. However, if there were a pronounced seasonality that substantially exceeded case-to-case variability, we would probably have noticed it. But it must be stressed that such constant distance thresholds don't easily capture all precipitation even within a given system -- let alone for different systems -- regardless of the season, which is part of the reason we opted for a two-threshold approach in order to focus on the precipitation close to the fronts while still capturing that at a greater distance as "far-frontal".

**12. Figure 2. This schematic implies that cyclonic and cold frontal precipitation are mutually exclusive. I guess this is not necessarily true, especially during the early stages of cyclone evolution. Also, given the cyclone is part of a larger-scale wave pattern, the location and shape of the high-pressure region in the schematic seems a little odd. What is the reasoning behind the shape and position of the high-pressure region in the schematic?**

Cyclonic and cold-frontal precipitation are, by our definition, indeed mutually exclusive. Of course, there is also precipitation which is simultaneously cold-frontal and cyclonic, and in principle we could further subdivide the cyclonic contributions into "purely cyclonic", "cyclonic/cold-frontal", etc. However, this would only further increase the number of components, which is already high enough at seven.

As for the schematic, it aims to represent some characteristics of high-pressure areas (as we defined them) as observed in our data and represented in the case studies (see figure below). In addition, the schematic shows the high-pressure region to overlap the far-frontal area, which highlight its precedence over the latter.

[Figure]

[Figure]
 caption continues below; the image reference is for the "That being said" figure.

That being said, we do agree that the shape of the high-pressure area in our schematic turned out somewhat peculiar, and we have therefore redrawn the schematic high-pressure area. In addition, we have extended the caption to highlight that the components are, indeed, mutually exclusive.

| Old | New |
|-----|-----|

*Old:*
*Figure 2.Schematic depiction of the seven front-cyclone-relative components high-pressure, cyclonic, cold-frontal, warm-frontal, collocated, far-frontal, and residual, as defined in Sec. 2.5.*

*New:*
*Figure 2.Schematic depiction of the seven front-cyclone-relative components high-pressure, cyclonic, cold-frontal, warm-frontal, collocated, far-frontal, and residual, as defined in Sec. 2.5. Note that they are mutually exclusive and cover the whole domain, i.e., at a given time step, each grid point is assigned to exactly one component.*

**13. Figure 3. This figure is too small to see the detailed frontal precipitation features.**

Since the reader can zoom in into the high-quality PDF, it should be possible for them to see the important features.

**14. Line 289. I do not see the warm front identified in figure 3b. If I understand correctly, this would be a red filled black contour. Where is this feature on the figure?**

The warm front is at this time step indeed not identified as a synoptic warm front, only as a local one (red contour). That's why we refer to it in the text merely as "a feature", which may be local or synoptic. "Warm front" in this sentence refers to what we know is there, not to what

the algorithm identifies (or doesn't). We concede that this sentence is not clear enough and have adapted it.

*Old:*
*The warm front east of the cyclone, now detected as a feature, is much weaker than the cold front and produces no precipitation, except close to the cold front, where occlusion may have commenced.*

*New:*
*The weak warm front east of the cyclone – now detected, albeit only as a local front – is much less pronounced than the cold front and produces no precipitation, except close to the cold front, where occlusion may have commenced.*

**15. Lines 295-300. In figures 3b and 3c there is a lot of precipitation that would generally be associated with the occluded/bent-back warm front which is not associated with frontal features using the objective criteria, nor within the cyclone feature contour. Which classification does this precipitation fall into? From figure 4 it looks to fall into the residual. This does not seem correct to me but is not referred to by the authors.**

In Fig. 3b, the whole bent-back portion of the precipitation band – actually most precipitation – is inside the cyclone contour and therefore classified as cyclonic. Note that we do mention that this missing front does not seem correct (l.295ff, "The precipitation band along its bent-back portion wraps almost completely around the cyclone center, much farther than the respective front feature, which suggests that not the whole front has been detected as a feature by our algorithm."), although in this case it would not make a difference as the cyclonic component takes precedence over the frontal ones in our algorithm.

In Fig. 3c, on the other hand, the remnants of the precipitation behind the cyclone center fall just outside the cyclone contour. However, there is a small cold-frontal feature east of Scotland, so at least some of this precipitation will be cold- and far-frontal. The southern part of this precipitation area presumably contributes to the residual precipitation feature in that area shown in Fig. 4h.

It is true that we do not explicitly refer to the residual precipitation in Fig. 3c. However, precipitation that should subjectively have been attributed to a cyclone or front but wasn't because the algorithm is not perfect is an inherent part of the residual component. Given the miss in Fig. 3c is, in our opinion, not egregious, we did not specifically comment on it. It would have been a completely different story, of course, if indeed the whole precipitation area bent around the cyclone center in Fig. 3b had been misclassified as residual; that definitely would require a comment.

**16. Figure 5. Similar to the comment above, in figure 5a there is a lot of precipitation close to the developing cyclone centre along a bent-back warm front. However, because this cyclone does not have a closed contour it is not captured by the cyclonic criteria. Would this just be assigned to the residual?**

In Fig. 5a, only the precipitation beyond 600 km from the outline of the warm front would be classified as residual, which likely captures most of this precipitation, so it will be classified as a

mixture of collocated, warm-frontal, and far-frontal. While there is indeed some residual precipitation in this area, as shown in Fig. 6h, that precipitation mostly stems from post-frontal precipitation and the secondary system visible around the British Isles in Fig. 5b and c.

**17. Line 327. What do the authors mean by the 'dry gap region between the fronts'? Is this the warm sector of the cyclone?**

No, this refers to the region between the tip of the cold front and the warm front that is oriented perpendicularly to it. However, it is indeed not well visible at the selected time steps – it would be more clearly visible in-between Figs. 5a and 5b. We have removed this sentence because it is indeed more confounding than helpful.

**18. Line 330. Browning and Roberts (1997) has a nice description of these cold frontal line features.**

Thank you for pointing this out, it is very interesting indeed. We have added a brief reference to that paper.

*Old:*
*In the cold sector behind the cyclone, there is widespread patchy precipitation, some of it associated with a relatively shallow cyclone near the British Isles.*

*New:*
*In the cold sector behind the cyclone, there is widespread patchy precipitation, some of it associated with a relatively shallow cyclone near the British Isles in a way reminiscent of secondary cold-frontal lines as described for instance by Browning et al. (1997).*

We fully agree and thank you for pointing this out. We have adapted the abstract accordingly.

*Old:*
*[...] The climatological analysis for the nine-year period shows that frontal precipitation peaks in fall and winter over the eastern North Atlantic, with cold frontal precipitation also being crucial year-round near the Alps; cyclonic precipitation is largest over the North Atlantic (especially in summer) and in the northern Mediterranean (except in summer); high-pressure precipitation occurs almost exclusively over land and primarily in summer; and the residual contributions uniformly amount to about 20 % in all seasons. Considering heavy precipitation events (defined based on the local 99.9$^{th}$ percentile) reveals that high-pressure precipitation dominates in summer over the continent; cold fronts produce much more heavy precipitation than warm fronts; and cyclones contribute substantially, especially in the Mediterranean in fall through spring and in Northern Europe in summer.*

*New:*
*[...] The climatological analysis for the nine-year period shows that frontal precipitation peaks in winter and fall over the eastern North Atlantic and the Alps (>70 % in winter), where cold frontal precipitation also being crucial year-round; cyclonic precipitation is largest over the North Atlantic (especially in summer with>40 %) and in the northern Mediterranean (widespread>40 %); high-pressure precipitation occurs almost exclusively over land and primarily in summer (widespread 30-60 %, locally>80 %); and the residual contributions uniformly amount to about 20 % in all seasons. Considering heavy precipitation events (defined based on the local 99.9$^{th}$ percentile) reveals that high-pressure precipitation dominates in summer over the continent (50–70 %, locally >80 %); cold fronts produce much more heavy precipitation than warm fronts; and cyclones contribute substantially (50–70 %), especially in the Mediterranean in fall through spring and in Northern Europe in summer.*

**22. Lines 420-425. The difference between the regions dominating heavy precipitation and overall precipitation is very interesting.**

Thank you, we agree!

**23. Line 436. Do the authors have a hypothesis for why cyclonic precipitation is not enhanced by topography in contrast to cold frontal precipitation?**

No, unfortunately we don't have a convincing explanation for this.

**24. Figure 7d. Does the lack of heavy precipitation associated with collocated fronts mean that ascent of warm conveyor belt over the warm front does not lead to heavy precipitation? This is surprising to me.**

We also expect that the ascent of WCBs over the warm front can lead to heavy precipitation, and Fig. 6d provides a nice example for this. Our assumption is that climatologically this ascent

over the warm front occurs more often over the identified warm fronts than the relatively small frontal segments classified as "collocated".

**25. Lines 495-508. This section is a repetition of your results and not a conclusion. I suggest removing this text.**

We agree that this section is not strictly necessary and have removed it.

*Old:*
*[...]*
*The meteorological results of the precipitation attribution show that different components are important in different geographical regions and in different seasons. When considering precipitation over the entire year, the most relevant weather systems are cold fronts near the Alps, warm fronts and cyclone centers in the North Atlantic and Western Europe, and cyclones in the Mediterranean, in particular near Italy and the Balkans. A substantial residual exists (about 20–30 %), indicating that our weather system categories do not encompass all precipitation-producing flow situations and that the attribution to the target systems is not perfect. Strong local enhancement occurs over high topography compared to the surrounding flat areas, 500 which is especially pronounced over the Alps and for cold-frontal precipitation. From a seasonal perspective, (i) cold fronts are important contributors in all seasons (especially over the continent), while warm fronts primarily contribute in winter and fall (especially over the North Atlantic); (ii) the largest cyclonic contributions shift from the Mediterranean in winter to Northern Europe in summer; and (iii) high-pressure precipitation is confined to summer over the continent, with pronounced local enhancement over the Alps. Focusing only on heavy-precipitation events reveals substantial differences to total precipitation: (i) Rather than over high-topography, heavy precipitation is particularly enhanced over land compared to sea; (ii) cold fronts also contribute substantially to heavy precipitation, whereas the relevance of warm fronts diminishes; (iii) cyclones are particularly important for heavy precipitation over the ocean; and (iv) the summertime high-pressure systems further gain in significance, in particular for continental summer convection. The results can be summarized concisely for several distinct geographical regions. [...]*

*New:*
*[...]*
*The meteorological results of the precipitation attribution can be summarized concisely for several distinct geographical regions. [...]*

**26. Lines 510-550. This section is interesting but should be strongly caveated by the fact that only 9-years of data has been used to create the climatologies. For example, there are many studies demonstrating decadal variability in the latitude of the storm track which would have a large influence on these conclusions.**

As regards summer precipitation events, previous studies suggest that the climatology is reasonably well captured by 10-year-long simulations (e.g. Ban et al. 2015, supplemental information), but for the winter seasons significant decadal variations of the NAO indicate that longer periods are indeed desirable or needed to compile a "real climatology".

To make this point, we have added the following sentences after the regional summary (line 559):

*New:*
*When summarizing these characteristics, it is important to mention another caveat: the comparatively short analysis period of nine years. While interannual variations in summer precipitation appear reasonably well covered with such simulations, variations in the North Atlantic oscillation suggest that longer integration*

*periods are desirable or needed in order to adequately cover decadal variations of the winter season. A significant challenge of such analyses is the costs of storing high-resolution output of multi-decadal simulations. It is thus desirable to use an online analysis approach that performs the respective analysis while the simulation is running (Di Girolamo et al., 2019; Schär et al., 2020) instead of storing all the relevant output data. Such an online analysis tool can also be highly beneficial when extending the feature-based analyses in three dimensions, e.g., by defining fronts in 3D and/or by considering the vertical structure of clouds and microphysical processes.*

[revised manuscript text omitted]

(a) total     (b) cold-frontal     (c) warm-frontal     (d) collocated

(e) high-pressure     (f) cyclonic     (g) far-frontal     (h) residual

1.0   1.5   2.2   3.2   4.6   7.0   10   15   22   32

precipitation (mm/day)

**Figure 6.** As Fig. 4, but for cyclone  Lancelot during the period  19–21 January 2007.

[Figure]

**Figure 7.** Mean daily precipitation during the nine-year period 2000–2008 (a) overall and (b–h) separated into seven front-cyclone-relative contributions.

[Figure]

**Figure 8.** Mean daily precipitation during (1–4) each season of the nine-year period 2000–2008 (a) overall and (b–e) of  select front-cyclone-relative contributions.

[Figure]

**Figure 9.** Relative precipitation contributions during (1–4) each season of the nine-year period 2000–2008 of front-cyclone-relative components: (a) sum of cold-frontal, warm-frontal, and collocated; (b) sum of cyclonic and far-frontal; (c) cyclonic; (d) high-pressure; and (ed) residual.

[Figure]

**Figure 10.** Like Fig. 7, but for annual heavy precipitation defined as the amount exceeding the local $99.9^{\text{th}}$ all-hour percentile of hourly precipitation intensity over the whole year.

[Figure]

**Figure 11.** Like Fig. 8, but for seasonal heavy precipitation defined as the amount exceeding the local $99.9^{\text{th}}$ all-hour percentile of hourly precipitation intensity in a given season.

[Figure]

**Figure 12.** Mean (a) total and (b) heavy precipitation over the analysis domain and six  select regions, as indicated in the map: British Isles, Western Europe, Alps, Southeastern Europe, Iberia, and the Mediterranean Sea. Heavy precipitation is defined as the amount of hourly precipitation above the local (grid-point specific) seasonal $99.9^{th}$ all-hour percentile. Each bar shows the annual-mean precipitation contribution of one front-cyclone-relative component (CF: cold-frontal; WF: warm-frontal; COL: collocated; FAR: far-frontal; CYC: cyclonic; HIP: high-pressure; RES: residual), with the four segments indicating the relative contribution of each season (DJF: winter; MAM: spring; JJA: summer; SON: fall). To obtain approximate absolute seasonal-mean amounts, multiply the height of a bar segment by four. Note that there is no relation between the colors of the bars and those of the regions on the map.

**Table 1.** Mid-monthly $|\nabla\theta_e|$ threshold values in $\mathrm{K}\,(100\,\mathrm{km})^{-1}$ to compute the thermal component of frontal areas, as described in Sec. 2.3.

| Jan | Feb | Mar | Apr | May | Jun | Jul | Aug | Sep | Oct | Nov | Dec |
|-----|-----|-----|-----|-----|-----|-----|-----|-----|-----|-----|-----|
| 4.0 | 4.0 | 5.0 | 6.0 | 7.0 | 8.0 | 8.0 | 8.0 | 7.0 | 6.0 | 5.0 | 4.0 |

**Table 2.** Mid-monthly $\Phi$ threshold values in $\mathrm{m}^2\mathrm{s}^{-2}$ to compute the $\Phi$-component of high-pressure areas at 850 hPa, as described in Sec. 2.4.

| Jan | Feb | Mar | Apr | May | Jun | Jul | Aug | Sep |
|-----|-----|-----|-----|-----|-----|-----|-----|-----|
|  1550 |  1550 |  1525 |  1500 |  1475 |  1450 |  1450 |  1450 |  1475 |  |

**Supplement to *"Attribution of precipitation to cyclones and fronts over Europe in a kilometer-scale regional climate simulation"**

Stefan Rüdisühli[1], Michael Sprenger[1], David Leutwyler[2], Christoph Schär[1], and Heini Wernli[1]

[1]Institute for Atmospheric and Climate Science, ETH Zurich, Switzerland
[2]Max Planck Institute for Meteorology, Hamburg, Germany

**Correspondence:** Stefan Rüdisühli (stefan.ruedisuehli@env.ethz.ch)

[Figure]

**Figure S1.** Feature frequencies of (a–d) cold fronts, (e–h) warm fronts, (i–l) cyclones, and (m–p) high-pressure areas during (left to right) winter, spring, summer, and fall 2000–2008. The outer black box shows the computational domain of the 2.2 km simulation, the inner box the analysis domain. The fields are computed by first reducing each feature in the respective time period to a binary mask field, and then averaging these binary fields to obtain the total feature frequency field.

[Figure]

**Figure S2.** Track frequencies of (a–d) cold fronts, (e–h) warm fronts, and (i–l) cyclones during (left to right) winter, spring, summer, and fall 2000–2008. The outer black box shows the computational domain of the 2.2 km simulation, the inner box the analysis domain. The fields are computed by first reducing each track to a binary mask field comprised of all grid points affected by any feature belonging to the track in the respective time period, and then averaging these binary fields to obtain the total track frequency field.

[Figure]

**Figure S3.** Frequencies of front-cyclone-relative  components during (0) the whole year, (1) winter (DJF), (2) spring (MAM), (3) summer (JJA), and (4) fall (SON) 2000–2008.  Shown are the (a) cold-frontal, (b) warm-frontal, (c) collocated, and (d) far-frontal components.

[Figure]

**Figure S4.**  Like Fig. S3, but showing the frequencies of the (e) cyclonic, (f) high-pressure, and (g) residual components.

[Figure]

**Figure S5.** Wet-hour frequency during (0) the whole year, (1) winter (DJF), (2) spring (MAM), (3) summer (JJA), and (4) fall (SON) 2000–2008, (a) overall and (b–e) for sets of front-cyclone-relative components, specifically: (b) sum of cold-frontal, warm-frontal, and collocated; (c) sum of cyclonic and far-frontal; (d) high-pressure; and (e) residual.

[Figure]

**Figure S6.** Like Fig. S5 b–d but for heavy precipitation, showing the frequency of hours with precipitation exceeding the local $99.9^{\text{th}}$ all-hour percentile of hourly precipitation.

---

## Author Response (AR2)

Paper wcd-2020-18

**Attribution of precipitation to cyclones and fronts over Europe in a kilometer-scale regional climate simulation**

Stefan Rüdisühli, Michael Sprenger, David Leutwyler, Christoph Schär, and Heini Wernli

**Response to the Reviewer's comments:**

We thank both reviewers for their very positive assessment of the revised version of our paper.

Below are the replies to the technical corrections suggested by reviewer #1.

**Reviewer #1**

I am very impressed how thoroughly the authors addressed the reviewers' comments. I find the revised paper to be much clearer.

It was a little confusing to see the tracked changes where all added text was crossed out. But the clean manuscript is fine.

I believe the manuscript can be accepted for publication in WCD.

We appreciate the reviewer's positive feedback and are delighted about their recommendation of acceptance. Furthermore, we apologize for the inconvenience caused by the wrong changes document; we accidentally mixed up the old and new documents when creating the diff.

**Minor remarks:**

l. 121: Are high-pressure systems also tracked?

No, they are not. In order to emphasize this, we have added a sentence at the end of Sec. 2.4 stating this explicitly:

*"In contrast to cyclones and fronts, high-pressure areas are not tracked over time."*

l. 245: This is a matter of personal preference, but I find 1/6 better than 0.167.

We agree that 1/6 is clearer, and indeed also what is used in the code. However, upon double-checking the value, we realized that the grouping criteria were slightly out of date, and have corrected them:

- *The* typical feature size *of a track is calculated by first combining, at each time step, the sizes of all features that belong to the track; and then calculating the*  *$80^{th}$ percentile of these total sizes over all time steps. Front tracks are only considered synoptic if the typical feature size is at least*  *400 grid points (~2000 km²).*
- *The* stationarity *of a track is determined as the typical feature size divided by the total footprint area (defined by all grid points that belong to the tracked front at any time). Front tracks are only considered synoptic if the stationarity is below*  *1/8.*

l. 564: *High-pressure heavy precipitation contributions are increased in summer and fall. While all other regions experience more high-pressure precipitation in summer than in fall, the opposite is true in the Mediterranean Sea.* I'd say something like : "The role of high-pressure systems in heavy precipitation increases in summer and even more so in fall. This is in contrast to all other regions where more precipitation occurs with high-pressure systems precipitation in summer than in fall".

We agree that these sentences could be formulated more clearly. We like the suggested alternative and have adopted it in slightly edited form:

*"* *The relevance of high-pressure systems for heavy precipitation increases in summer and even more so in fall.*  *This is in contrast to all other regions, where more heavy precipitation is associated with high-pressure areas in summer than in fall."*

l. 575: While I agree that analysis of a longer period may be insightful, I do not get the reasoning within these lines: Why NAO variability suggests that a longer period is needed? Did the nine years of the analysis see only a narrow range of possible NAO indexes?

Yes, we wanted to express that with 9 winters, we most likely cannot represent the full variability of the larger-scale North Atlantic - European circulation, as expressed, e.g., by the NAO.

However, we agree the mentioning the NAO at this place might be a bit too specific. We therefore simplified the sentence as follows:

*When summarizing these characteristics, it is important to mention another caveat: the comparatively short analysis period of nine years. While interannual variations in summer precipitation appear reasonably well covered with such simulations,  nine years might not be enough to fully capture the high variability of the large-scale atmospheric flow that determines European weather conditions in winter.*

l. 578: "Such an online analysis ..." I think 'an' is not needed.

We disagree: As far as we can tell, the "an" is indeed needed (unless "tool" is pluralized). However, given the tool in question is merely implied in the preceding sentence, we have adapted the sentence:

*"Such an  approach can also be highly beneficial [...]."*

Fig.12: Similar to WF and CF, I'd refer to high-pressure systems as HP (but still like 'FAR' for the far frontal). Instead of "… the annual-mean precipitation contribution of one front-cyclone-relative component…", I'd say "… the seasonal precipitation contribution from each of seven components (or weather systems) …"

While we agree that "HP" may be the slightly better choice than "HIP", we think that both choices are OK and this change thus not worth the effort of reproducing the whole figure.

Regarding the caption, we stick with "annual-mean" over "seasonal" because the latter would imply that the height of the four seasonal segments corresponds to the amount of precipitation in that season on the vertical axis. The vertical axis, however, shows the annual precipitation amounts. We concede that the formulation is complicated, but in this case, correctness trumps elegance. For the same reason, we stick with "front-cyclone-relative component" over "seven components/weather systems".

l. 542: Instead of 'They stand out' I suggest "This region stands out"

We agree and have change the text as proposed.

l. 550: I'd change to something like "…, precipitation in Southern Europe benefits greatly form cyclones, while the contribution from warm fronts is reduced. The latter is observed in all southern regions of the domain"

We like the suggested alternative and have adopted it in slightly edited form:

" As over the British Isles,  precipitation in Southeastern Europe *benefits greatly from cyclones*,  *while the* warm-frontal  *contributions are reduced. The latter is observed in all southern regions of the domain.*"

As a final remark, I think that residual precipitation may be associated with high-level processes (such as, e.g., cut-off lows) that cannot be identified within the weather systems considered in the paper. This can explain a large amount of unattributable precipitation over the Iberian Peninsula.

We agree with this hypothesis and have added a respective remark to the discussion of the IP in the Conclusions:

*"[...] The fraction of unattributable precipitation is large compared with other regions, especially in spring, which may be partially explained by the prevalence of upper-level cut-off lows (e.g., Nieto et al., 2007)."*

*Nieto R, Gimeno L, Añel JA, de la Torre L, Gallego D, Barriopedro D, Gallego M, Gordillo A, Redaño A, Delgado G. 2007. Analysis of the precipitation and cloudiness associated with COLs occurrence in the Iberian Peninsula. Meteorol. Atmos. Phys. 96: 103–119.*

I'd appreciate if you could comment why the Mediterranean sees a lot of precipitation from cyclones compared to fronts: is it because fronts are shorter and do not extend much outside the cyclone area or because there are many non-frontal cyclones (if so, do they mainly develop from the upper troposphere, e.g. following a Rossby wave breaking, or from the surface due to local heating. The former may explain why cyclones play even larger role in heavy precipitation). I understand that analysis of the development and structure of cyclones is outside the focus of the paper, but some ideas either from your experience or other papers would be valuable.

We agree with all these hypotheses of why cyclones dominate precipitation in the Mediterranean. First, the dynamics of Mediterranean cyclones is typically governed by narrow streamers of upper-level stratospheric PV, which result from Rossby wave breaking, and their interaction with latent heating in the center of the evolving cyclones (e.g., Flaounas et al., 2015). Precipitation is therefore often most intense close to the cyclone center (see Fig. 7 in Flaounas et al., 2015). Also, Mediterranean cyclones are typically smaller than North Atlantic cyclones and fronts are shorter and less intense. The reasons for this are that Mediterranean cyclones are instigated by narrow

troughs (PV streamers) compared to the broader troughs over the North Atlantic, and they evolve in a spatially confined ocean basin, such that, e.g., cold fronts cannot extend far equatorward from the cyclone centers.

This is corroborated by the feature and feature track frequency composites in Figs. S1 and S2, respectively, which both show that the decrease in frequency over the Mediterranean compared to the North Atlantic is more pronounced for fronts than for cyclones. Given that the composite frequency is dependent on the feature size in addition to their occurrence frequency, this supports the hypothesis that fronts are smaller with respect to cyclones in the Mediterranean than over the North Atlantic, and that therefore more precipitation along those fronts falls within the boundaries of the associated cyclones.

We have added a reference to Flaounas et al. (2015) to the discussion of the Mediterranean in the Conclusions:

[revised manuscript text omitted]

(a) total    (b) cold-frontal    (c) warm-frontal    (d) collocated

(e) high-pressure    (f) cyclonic    (g) far-frontal    (h) residual

.10   .20   .30   .40   .50

[a] precipitation (mm / d)

.0500   .100   .150   .200   .250

[b–h] precipitation (mm / d)

**Figure 10.** Like Fig. 7, but for annual heavy precipitation defined as the amount exceeding the local $99.9^{\text{th}}$ all-hour percentile of hourly precipitation intensity over the whole year.

[Figure]

**Figure 11.** Like Fig. 8, but for seasonal heavy precipitation defined as the amount exceeding the local $99.9^{\text{th}}$ all-hour percentile of hourly precipitation intensity in a given season.

[Figure]

**Figure 12.** Mean (a) total and (b) heavy precipitation over the analysis domain and six selected regions, as indicated in the map: British Isles, Western Europe, Alps, Southeastern Europe, Iberia, and the Mediterranean Sea. Heavy precipitation is defined as the amount of hourly precipitation above the local (grid-point specific) seasonal $99.9^{\text{th}}$ all-hour percentile. Each bar shows the annual-mean precipitation contribution of one front-cyclone-relative component (CF: cold-frontal; WF: warm-frontal; COL: collocated; FAR: far-frontal; CYC: cyclonic; HIP: high-pressure; RES: residual), with the four segments indicating the relative contribution of each season (DJF: winter; MAM: spring; JJA: summer; SON: fall). To obtain approximate absolute seasonal-mean amounts, multiply the height of a bar segment by four. Note that there is no relation between the colors of the bars and those of the regions on the map.

**Table 1.** Mid-monthly $|\nabla\theta_e|$ threshold values in $\mathrm{K\,(100\,km)^{-1}}$ to compute the thermal component of frontal areas, as described in Sec. 2.3.

| Jan | Feb | Mar | Apr | May | Jun | Jul | Aug | Sep | Oct | Nov | Dec |
|-----|-----|-----|-----|-----|-----|-----|-----|-----|-----|-----|-----|
| 4.0 | 4.0 | 5.0 | 6.0 | 7.0 | 8.0 | 8.0 | 8.0 | 7.0 | 6.0 | 5.0 | 4.0 |

**Table 2.** Mid-monthly $\Phi$ threshold values in $\mathrm{m^2\,s^{-2}}$ to compute the $\Phi$-component of high-pressure areas at $850\,\mathrm{hPa}$, as described in Sec. 2.4.

| Jan | Feb | Mar | Apr | May | Jun | Jul | Aug | Sep | Oct | Nov | Dec |
|-----|-----|-----|-----|-----|-----|-----|-----|-----|-----|-----|-----|
| 15,500 | 15,500 | 15,250 | 15,000 | 14,750 | 14,500 | 14,500 | 14,500 | 14,750 | 15,000 | 15,250 | 15,500 |